# Therapeutic 6-thio-deoxyguanosine inhibits telomere elongation in cancer cells by inducing a non-productive stalled telomerase complex

Samantha L. Sanford[1], Mareike Badstübner[2], Michael Gerber[1], William Mannherz [3,4,5], Noah Lampl[4,5], Rachel Dannenberg[6], Angela Hinchie[7,8], Matthew A. Schaich[1,9], Sua Myong [10,11], Mark Hedglin [6], Suneet Agarwal [3,4,5], Jonathan K. Alder [7,8], Michael D. Stone [2,12] & Patricia L. Opresko [1,9,13] ✉

Most cancers upregulate the telomere lengthening enzyme telomerase to achieve unlimited cell division. How chemotherapeutic nucleoside 6-thio-2'-deoxyguanosine (6-thio-dG) targets telomerase to inhibit telomere maintenance in cancer cells and tumors was unclear. Here, we demonstrate that telomerase insertion of 6-thio-dGTP prevents synthesis of additional telomeric repeats but does not disrupt telomerase binding to telomeres. Specifically, 6-thio-dG inhibits telomere extension after telomerase translocates along its product DNA to reposition the template, inducing a non-productive complex rather than enzyme dissociation. Furthermore, we provide direct evidence that 6-thio-dG treatment inhibits telomere synthesis by telomerase in cancer cells. In agreement, telomerase-expressing cancer cells harboring critically short telomeres are more sensitive to 6-thio-dG and show a greater induction of telomere losses compared to cancer cells with long telomere reserves. Our studies reveal that telomere length and telomerase status determine 6-thio-dG sensitivity and uncover the molecular mechanism by which 6-thio-dG selectively inhibits telomerase synthesis of telomeric DNA.

Telomeres are nucleoprotein DNA structures that protect the ends of linear chromosomes and are essential for genome stability and cellular proliferation. Human telomeres consist of approximately 10-15 kilobases (kb) of tandem double-stranded GGTTAG repeats, a 3'-single-stranded (ss) G-rich overhang ranging from 50 to 200 nt, and a complex of six proteins that are collectively called Shelterin[1]. Telomeres progressively shorten with each cell division, and extensive telomere shortening or losses trigger cell senescence and aging-related pathologies[2,3]. Critically short telomeres are falsely recognized as chromosome breaks, thereby activating an ATM and/or ATR kinase-mediated DNA damage response (DDR), leading to p53 activation[1].

Dysfunctional telomeres are indicated by the localization of DDR factors 53BP1 or phosphorylated histone H2AX to telomeres, termed TIF or DDR+ telomeres[2,4]. In this context, dysfunctional telomeres serve as an initial barrier to tumorigenesis. However, when p53 pathways are compromised or mutated, cells may bypass senescence and continue to divide with critically short telomeres[5]. During this period, DNA double-strand break repair (DSBR) mechanisms generate chromosome end-to-end fusions, thereby promoting chromosomal instability that kills most cells but drives malignant transformation in those that survive[6,7]. The surviving cells either activate the recombination-based alternative lengthening of telomeres (ALT) pathway or, in most cases,

---

upregulate telomerase to maintain telomeres to enable unlimited cell proliferation[8,9].

Telomerase has been identified as a target for cancer therapy since it is expressed in over 85% of cancers[10], and most normal somatic cells lack telomerase[3,11]. Cancer cells typically undergo extensive telomere shortening before telomerase upregulation and generally maintain their telomeres at shorter lengths than normal cells[8,12]. Thus, transient telomerase inhibition has the potential to deplete telomeres more rapidly in highly proliferating cancer cells with short telomeres while reducing the risk for extensive telomere shortening in non-cancer cells. Telomerase has a catalytic reverse transcriptase subunit (hTERT) and a functional RNA (hTR) that harbors the template sequence for adding new telomeric repeats[13]. Somatic mutations in the proximal hTERT promoter are among the most common in non-coding regions within cancer genomes, and are particularly frequent in melanomas, glioblastomas, liposarcomas, and urothelial cancers[14,15]. These mutations reactivate telomerase by generating a novel ETS transcription factor binding site, that contributes to telomerase upregulation[16]. The recently FDA-approved telomerase inhibitor Imetelstat, an antisense phosphorothioate oligonucleotide that targets the telomerase RNA template, has shown some clinical efficacy in reducing myelodysplastic and myeloproliferative neoplasms[17–19]. However, potential off-target effects of antisense therapies may contribute to this drug's anti-tumor effects and/or toxicity[20,21].

The unique catalytic properties of telomerase, when compared with other DNA polymerases, led to the search for nucleoside/nucleotide analogs that could selectively target telomerase[22]. Thiopurines, including 6-thioguanine and 6-mercaptopurines, are used therapeutically as antileukemic, anti-inflammatory, and immunosuppressive agents[23]. High toxicity has limited their use to leukemia and some pediatric cancers[24]. Thiopurines are metabolized to 6-thio-GMP and then to 6-thio-GTP, which can inhibit GTPases and purine biosynthesis, or to 6-thio-dGTP, which can be incorporated into DNA[23,25]. However, the Shay lab showed the 6-thio-dG deoxyribose analog is less toxic to mice and more effectively decreases the growth of A549 lung tumors in mouse xenograft studies, compared to the 6-thio-G drug[26]. Treatment with 6-thio-dG, now called THIO, promotes telomere shortening and dysfunction (indicated by DDR+ telomeres) in telomerase-expressing cancer cells, and is less toxic to telomerase-deficient non-diseased cells, suggesting telomerase insertion of 6-thio-dGTP impairs telomere maintenance[26]. Subsequent preclinical studies show 6-thio-dG reduces tumor growth and promotes telomere dysfunction in melanoma, lung, gliomas and pediatric brain xenograft tumors in mice[27–32]. Furthermore, 6-thio-dG increases DDR+ telomeres in colorectal, lung and hepatocellular carcinomas from syngeneic mouse models and synergizes with immune checkpoint inhibitors to reduce tumor growth[33,34]. These studies provide evidence that 6-thio-dG targets telomerase and is a promising anti-cancer therapeutic in combination with other chemotherapeutics or immunotherapy. However, the mechanism of 6-thio-dG action and specificity for telomerase remained poorly understood.

We previously showed that human telomerase can readily add 6-thio-dGTP to telomere DNA in vitro, but insertion strongly inhibits further repeat additions by an unknown mechanism[35]. The telomerase catalytic cycle begins when the telomeric single-strand DNA (ssDNA) overhang base pairs with the complementary region of the telomerase RNA template (Fig. 1a), comprising a 4-5 nucleotide (nt) alignment region and 6 nt templating bases[36]. After incorporating an incoming dNTP, the active site moves to the next template base to insert a new dNTP. Telomere elongation continues to the 5' template boundary, referred to as nucleotide addition processivity (NAP), which is not blocked by 6-thio-dGTP[35]. Then telomerase can dissociate or translocate on the DNA product to realign the template for repeat addition processivity (RAP)[37]. Of the various oxidatively damaged and therapeutic dNTPs we tested biochemically, 6-thio-dGTP exhibited the

lowest $IC_{50}$ defined as the concentration required to reduce RAP by half[35]. Single-nucleotide insertion kinetic studies of an hTERT catalytic core homolog, *Tribolium castaneum* (tcTERT), revealed that the catalytic efficiency of 6-thio-dGTP is remarkably similar to dGTP, indicating poor selectivity against this therapeutic nucleotide[35]. These in vitro results show telomerase readily inserts 6-thio-dGTP, but this impairs further cycling to add more repeats.

Here, we examine how 6-thio-dGTP insertion by telomerase halts further repeat addition in vitro, and investigate how 6-thio-dG causes telomere shortening and dysfunction in human cancer cells. We find telomeric DNA with a terminal, pre-existing 6-thio-dG, mimicking catalytic addition of 6-thio-dGTP, are poorly extended by telomerase. However, the presence of 6-thio-dG in the telomere does not impair telomerase binding. Rather, we provide evidence that the catalytic addition of 6-thio-dG causes telomerase to enter a non-productive telomere-bound state, thereby preventing further extension after translocation, without promoting telomerase dissociation from the telomere. We further demonstrate that 6-thio-dG treatment of human cancer cells inhibits telomeric repeat synthesis by telomerase and impairs telomere stability. Moreover, we demonstrate that telomerase status and telomere length in cancer cells determine 6-thio-dG sensitivity. Our results reveal that 6-thio-dG inhibits telomere maintenance in cancer cells by producing a dysfunctional telomerase complex bound to the telomere following the addition of 6-thio-dGTP and enzyme translocation.

## Results

### Telomerase insertion of 6-thio-dGTP strongly inhibits repeat addition processivity

Previously, we found that substituting dGTP for 6-thio-dGTP in telomerase reactions limits telomere extension to one repeat[35]. Here, we asked if telomerase processivity factors POT1 and TPP1 increase the 6-thio-dGTP $IC_{50}$ and, therefore, potentially reduce its potency as a telomerase inhibitor in cells when Shelterin is present. For this, we isolated overexpressed FLAG-tagged telomerase from human cells (Supplementary Fig. 1a). Telomerase primer extension reactions were conducted with $^{32}$P-end labeled DNA primer, cellular relevant concentrations of natural dNTPs (5 μM dGTP, 24 μM dATP, 29 μM dCTP, and 37 μM dTTP)[38] and increasing 6-thio-dGTP concentration from 0 to 100 μM. We used a primer with a single C substitution to position POT1-TPP1 at the 5' end of the primer (Fig. 1b). Telomerase processivity decreased with increasing 6-thio-dGTP concentration with an $IC_{50}$ of $3.2 ± 2.5$ μM (Fig. 1b, lanes 1–5, Fig. 1c, d and Supplementary Fig. 1d), consistent with our previous result[35]. POT1-TPP1 addition (Fig. 1b, lane 6) increased processivity compared to the control (lane 1), confirming telomerase stimulation. However, increasing 6-thio-dGTP concentrations (Fig. 1b, lanes 7–10, and Fig. 1c, d) strongly reduced telomerase processivity, with an $IC_{50}$ value of $7.5 ± 0.1$ μM even in the presence of POT1-TPP1. Notably, POT1-TPP1 stimulation is more obvious in reactions conducted with $^{32}$P-dGTP instead of $^{32}$P-end labeled primers, because long products have more $^{32}$P-dG incorporated compared to short products[39]. Therefore, we also compared the proportion of short (1-4 repeats added) to long (≥ 4 repeats added) products. For end-labeled primers, this analysis more clearly demonstrates POT1-TPP1 stimulation of processivity and confirms lack of rescue from 6-thio-dGTP inhibition (Supplementary Fig. 1c). These data suggest that micromolar 6-thio-dG drug has the potential to strongly inhibit telomerase in cells when telomerase accessory factors POT1 and TPP1 are present.

We next confirmed that 6-thio-dGTP inhibition of telomerase processivity was not due to reduced catalysis or disruption of G-quadruplex (G4) folding in the product DNA[40]. Time course reactions reveal similar percent primer extension over time for adding a single dGTP compared to 6-thio-dGTP (Supplementary Fig. 1e), consistent with our previous pre-steady-state kinetic study using tcTERT[35].

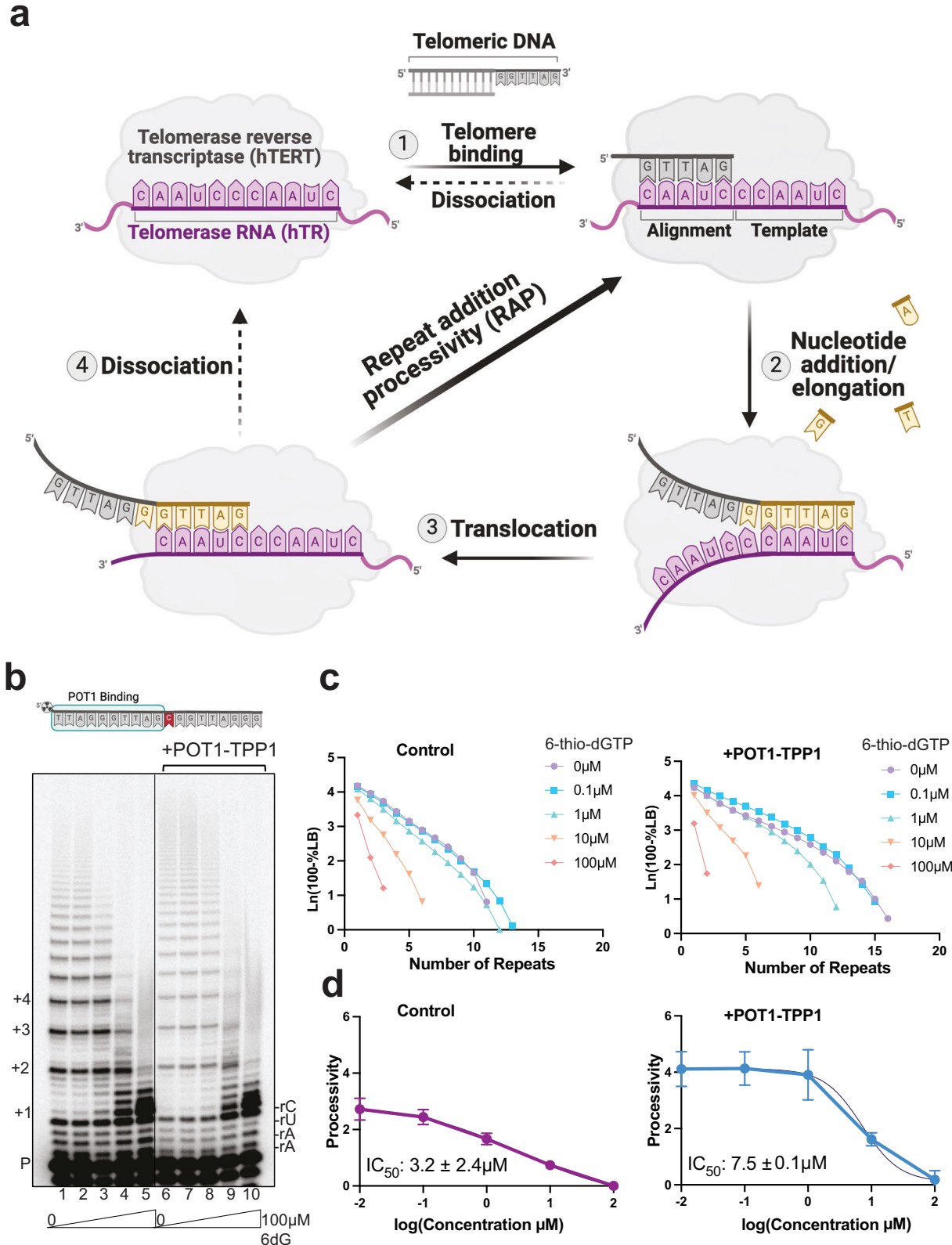

Furthermore, replacing the G4 stabilizing cation K⁺ with Li⁺, which cannot stabilize G4s[41], fails to rescue or suppress 6-thio-dGTP inhibition of telomerase (Supplementary Fig. 1f, g).

Next, we asked if 6-thio-dGTP is a telomerase-specific inhibitor. For this, we conducted reactions with fully reconstituted human replicative DNA polymerase δ (Pol δ) holoenzyme along with its processivity factor PCNA and dNTPs within physiological concentrations

(see Methods)[38]. This assay assembles and stabilizes a single PCNA onto all DNA substrates, and monitors DNA synthesis from a single encounter of Pol δ with a DNA substrate to measure nucleotide addition processivity (NAP)[42] (Fig. 2a, b and Supplementary Fig. 2a). The substitution of 6-thio-dGTP for dGTP did not significantly alter primer extension, although the percent of full-length product decreased somewhat (Fig. 2c). To further examine how 6-thio-dGTP impacts NAP,

**Fig. 1 | Telomerase processivity factors POT1-TPP1 cannot rescue 6-thio-dGTP inhibition of telomerase RAP. a** Cartoon of the telomerase catalytic cycle. The numbers represent each step in the cycle. Gray indicates the telomeric DNA primer (see Supplementary Table 1, primer 1). Purple indicates the telomerase RNA template and hTERT. Yellow indicates the newly added nucleotides. Created in BioRender. Sanford, S. (2025) https://BioRender.com/90uszs7. **b** Direct telomerase assays were conducted in the absence or presence of 500 nM POT1 and 500 nM TPP1, as indicated, with physiologic dNTP concentrations (24 µM dATP, 29 µM dCTP, 37 µM dTTP, 5.2 µM dGTP) and 5 nM $^{32}$P-end labeled primer (TTAGGGGTTAGCGTTAGGG) designed to position POT1 at the 10 nt primer 5′ end. Reactions contained 0–100 µM 6-thio-dGTP. Numbers on the left indicate the number of added repeats, letters on the right indicate template base, and P indicates unextended 18-mer primer. Created in BioRender. Sanford, S. (2025) https://BioRender.com/igvfg46. **c** Processivity (R 1/2) calculated on the basis of total products normalized to loading control shown for replicate 1 (see Supplementary Fig. 1d for replicates 2 and 3). **d** IC$_{50}$ values calculated from processivity (R 1/2). Data represent and mean and s.d. from 3 independent replicates. Source data are provided as a Source Data file.

we measured the polymerase probability of insertion (Pi) at each template position. Pi represents the probability that Pol δ will processively add another dNTP after catalysis versus dissociating from the DNA substrates, and therefore lower Pi values indicate the enzyme is more likely to stall and dissociate. 6-thio-dGTP substitution did not significantly alter the Pi values for most template positions, except for the 15th dNTP and 32$^{nd}$ dNTP insertion steps which both reflect insertion after extending from a 6-thio-dG-C base pair (Supplementary Fig. 2b). Our results agree with earlier reports that 6-thio-dGTP does not inhibit DNA polymerases β, α, γ and δ[43,44], however, we cannot rule out that other DNA polymerases may be impacted. Nevertheless, while 6-thio-dGTP substitution decreases the progression of replicating Pol δ holoenzyme slightly, this impact is minor compared to the strong inhibition of telomerase.

## Telomerase retains binding to 6-thio-dG containing telomeric DNA

Telomerase differs from DNA polymerases in that synthesis after repeat addition requires the enzyme to reposition the template while retaining interaction with its DNA product. During repeat synthesis, the telomerase RNA-DNA hybrid is maintained at 4 or 5-bp; the optimal size for active site accommodation[36,45]. The hybrid melts when telomerase translocates to form a new RNA-DNA hybrid and regenerates the template. We predicted that telomerase addition of 6-thio-dG may disrupt hybrid formation and cause telomerase dissociation after translocation. To test this, we used a series of 18-mer telomeric DNA substrates containing a 6-thio-dG at different positions in the last five nucleotides, corresponding to the template alignment region and mimicking 6-thio-dG insertion (Fig. 3c, (GGTTAG)$_2$ followed by sequences 1-7). We conducted telomerase activity assays with 5′-radiolabeled primers and compared percent primer extension. Placement of the 6-thio-dG at, or proximal to, the primer 3′ end reduced telomerase activity (Fig. 3, compare substrate 1 with 2, and substrate 4 with 5 and 6). Since telomerase can add two consecutive 6-thio-dGTPs prior to translocation[35], we tested a primer with two terminal 6-thio-dGs and found this strongly reduced telomerase activity (Fig. 3, substrate 7). However, moving the 6-thio-dG further from the 3′ end rescues telomerase inhibition (Fig. 3, compare substrate lane 1 with 3). Thus, the ability of 6-thio-dG to inhibit telomerase and potentially impair re-annealing with the alignment region depends on its position in the primer substrate.

Since a 6-thio-dG within the RNA-DNA hybrid more strongly inhibited telomerase compared to a 6-thio-dG outside the annealing region, we predicted the modified base might impair telomerase binding to the DNA. To test this, we conducted electrophoretic mobility shift assays (EMSAs) with 1 to 20 nM Halo-3xFLAG-tagged telomerase (Halo-telomerase) and 2.5 nM fluorescently labeled telomeric DNA with and without 6-thio-dG at the 3′ end (Fig. 3d). Halo-telomerase conjugated with photostable Janella fluorophore (JF-635) allows visualization of telomerase but does not impair telomerase activity (Supplementary Fig. 1a)[46]. 6-thio-dG did not significantly alter the percent DNA bound to the unmodified telomere DNA (Fig. 3d, e, and Supplementary Fig. 3b, c). Telomerase complex that failed to enter the gel may represent an improperly folded or assembled complex, or hTERT lacking hTR, and is observed even with a non-telomeric substrate that shows very weak binding to telomerase (Fig. 3d and Supplementary Fig. 3a). To confirm the EMSA results, we also examined 6-thio-dG impact on telomerase binding at the single molecule level using a substrate that mimics the telomeric overhang extending from dsDNA tethered between two optically-trapped beads[47]. Single-molecule binding dynamics of Halo-telomerase were recorded in real-time with a C-trap correlative optical tweezers and fluorescence microscope. For both substrates with and without 6-thio-dG, we observed events in which telomerase bound very briefly (1-10 s), and events in which telomerase remained bound to the telomeric overhang for long time periods greater than 45 seconds (Supplementary Fig. 3e). Consistent with the EMSA results, telomerase did not bind an overhang lacking telomere repeats (Supplementary Fig. 3g) and a terminal 6-thio-dG did not significantly reduce the average telomerase dwell time, but rather slightly increased this value. Modeling short and long lifetimes as cumulative residence time distributions and fitting to a double exponential decay function yielded average lifetimes ranging from 48 s for the natural telomere sequence to 64 s when 6-thio-dG was present (Supplementary Fig. 3f). Collectively, these results confirm that a pre-existing 6-thio-dG does not impair telomerase binding to the telomeric DNA.

## Telomerase 6-thio-dGTP addition inhibits repeat addition processivity but induces DNA product dynamics

Results from the binding assays above suggest that 6-thio-dG insertion does not inhibit telomerase RAP by impairing enzyme binding to the DNA. To further explore the impact of 6-thio-dG on telomerase catalysis, we used single-molecule analysis to directly visualize how the incorporation of 6-thio-dGTP impacts the dynamics of telomere DNA product bound to telomerase enzyme. We reconstituted the human telomerase catalytic core using a cell-free translation system[48] with an ultrastable Lumindyne Dye-555 (LD555) at position U42 of hTR, proximal to the active site, which we previously confirmed has catalytic activity comparable to endogenous telomerase[49,50]. We used a single-molecule Förster Resonance Energy Transfer (FRET) assay to report on the energy transfer, and therefore the distance, between the donor dye (LD555) on hTR, and an acceptor dye (Cy5) on the (TTAGGG)$_3$ substrate (Fig. 4a and Supplementary Table 1). The ability to visualize proximity between the two dyes reports on RNA-DNA dynamics during telomerase activity in real-time within individual telomerase-DNA complexes.

Telomerase was pre-incubated with the DNA substrate before immobilizing complexes via a 5′-biotin on a quartz slide for smFRET measurements. The FRET histograms showed telomerase stably bound to the substrate produced a high FRET peak centered at ~ 0.7 (Fig. 4, top panels, "stalled"), consistent with previous reports[50]. Adding natural dNTPs generated the expected shift to broad peaks with a new mid-FRET peak and eventually to a low FRET peak at ~ 0.4 after 30 min. The time-dependent shift toward lower FRET values reports on telomerase movement away from the acceptor dye on the DNA substrate along the newly synthesized DNA product (Fig. 4a, c). Importantly, incubating stalled telomerase-DNA complexes for 30 min in our assay conditions lacking dNTPs showed no shift in the smFRET distribution,

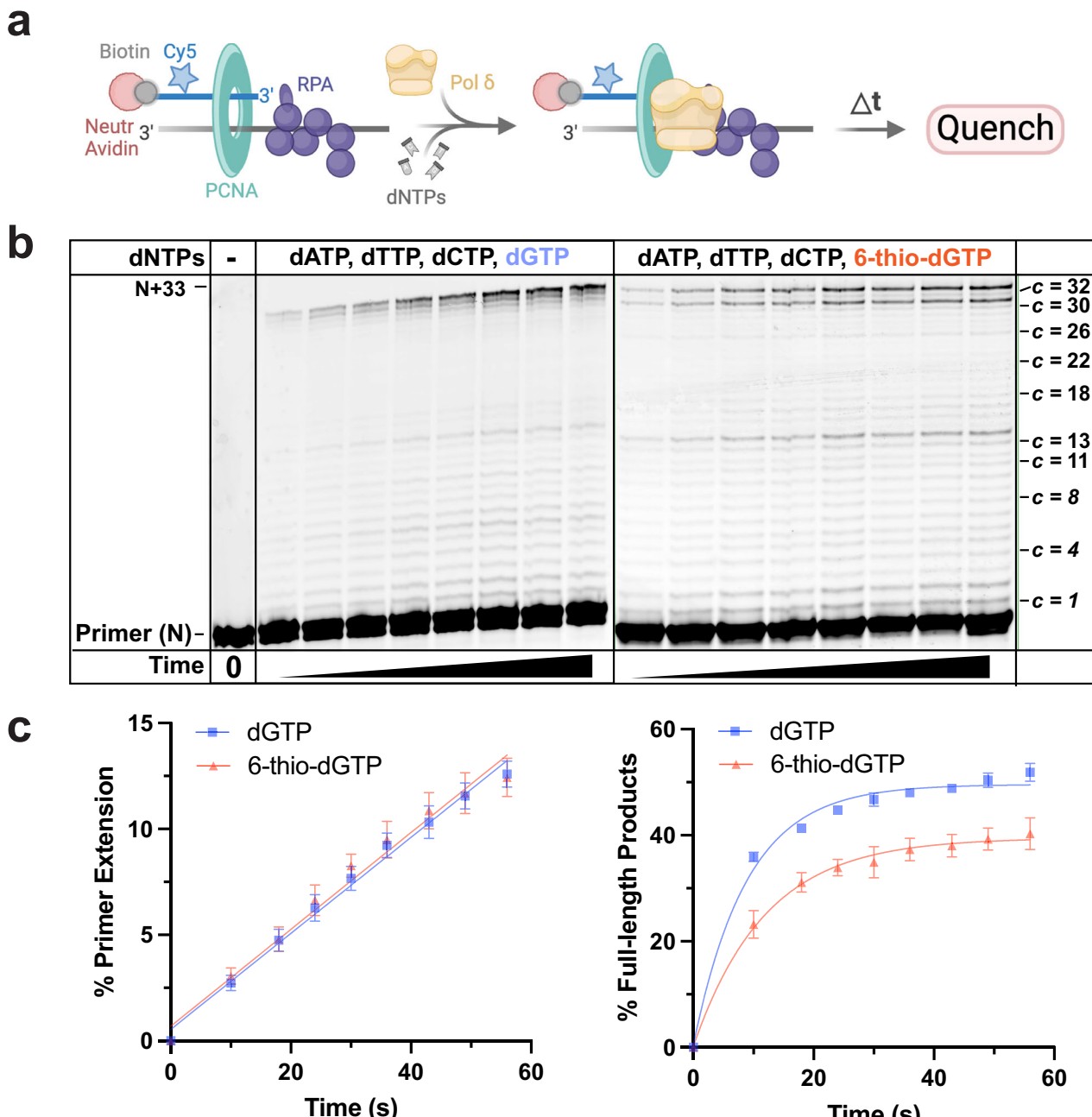

**Fig. 2 | 6-thio-dGTP fails to inhibit replicative DNA polymerase δ holoenzyme.**
**a** Schematic of the assay to monitor primer extension by Pol δ holoenzymes during a single binding encounter with a DNA substrate (BioCy5P/T, Supplementary Fig. 2a). PCNA (green) is assembled onto the substrate (250 nM), orientated toward the primer/template junction. Following the addition of physiological dNTP concentrations, DNA synthesis is initiated by adding 8.8 nM Pol δ. Created in BioRender. Sanford, S. (2025) https://BioRender.com/6ab6ej1. **b** Representative 16% denaturing gel of the primer extension products. The sizes of the primer (N) and the full-length product (N + 33) are indicated on the left, and the dNTP insertion step at each template C (c) is indicated on the right. **c** Quantification of DNA synthesis. The left panel shows the percent total primer extension plotted as a function of time. Data points after time = 0 s are fit to a linear regression. The right panel shows the percent of full-length products plotted as a function of reaction time. Data from reactions with natural dNTPs are shown in blue, and data from reactions in which 6-thio-dGTP replaced dGTP are shown in orange. Each data point represents the mean ± S.E.M. of three independent reactions. Source data are provided as a Source Data file.

supporting the conclusion that any deviation from the initial stalled population at ~0.7 FRET is telomerase catalysis dependent (Fig. 4b). The small subpopulation remaining in the high FRET state after 30 min may represent catalytically inactive enzyme or DNA-telomerase complexes not primed for activity (Fig. 4c). In contrast, the FRET histograms from experiments with 6-thio-dGTP substituted for dGTP (Fig. 4d), or a 10-fold excess of 6-thio-dGTP over dGTP (Fig. 4e) failed to produce the

expected drop in FRET and mid-FRET peak even after 30 min reaction. This is consistent with a reduction of processive telomere DNA repeat synthesis as shown in our bulk primer extension assays under similar conditions (Figs. 1b–d and 3a–c). We reasoned appearance of the small, low FRET peak (~0.2) likely reflects transient dynamics and sampling of lower FRET states (Fig. 5) caused by completing synthesis of the first repeat, consistent with the bulk biochemistry experiments. Results with

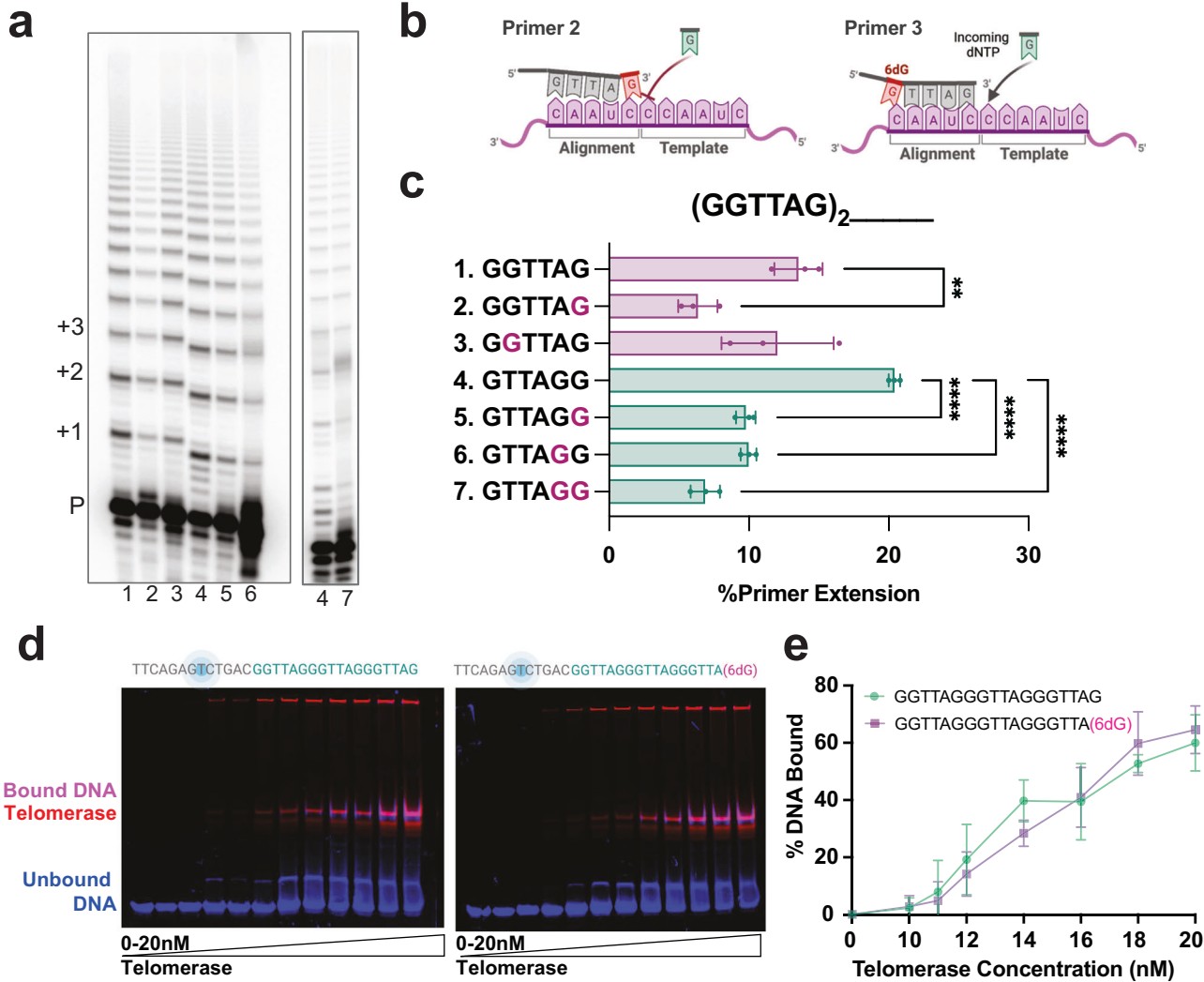

**Fig. 3 | Terminal 6-thio-dG bases impair telomerase RAP but not telomerase binding. a** Telomerase reactions conducted with 6-thio-dG at different positions on the telomeric substrate primer. P marks the unextended 18-mer primer, and numbers indicate the number of added repeats. Lane number corresponds to primer sequence in panel (**c**). **b** Cartoons of telomeric primer (gray) with 6-thio-dG (red), telomerase RNA template (purple), and incoming dNTP (green). Created in BioRender. Sanford, S. (2025) https://BioRender.com/brfbsrx. **c** Quantitation of percent primer extension. Data represent the mean ± s.d. from three independent experiments. Statistical significance was determined by one-way ANOVA (**$P = 0.0037$; ****$P < 0.0001$). Sequence of 3′ terminal telomeric repeat in the 18 mer primer following (GGTTAG)$_2$ is shown on the Y axis (see Supplementary Table 1). Purple G's indicate 6-thio-dG modification. **d** Binding reactions were conducted with 2.5 nM fluorescein labeled telomere substrate (30 mer) with a 3′ terminal dG or 6-thio-dG (blue) (see Supplementary Table 1) and 1–20 nM Halo-telomerase conjugated with JF-635 dye (red), for 30 min at room temperature, and separated by EMSA to visualize telomerase-bound substrate (pink). **e** Quantification of % DNA bound versus estimated Halo-telomerase concentration. Data represent the mean ± s.d. from three independent experiments. Source data are provided as a Source Data file.

6-thio-dGTP excess showed a slightly greater shift to mid FRET (between ~ 0.4–0.6), compared to 6-thio-dGTP replacing dGTP, as expected since dGTP can compete with 6-thio-dGTP (Fig. 4d, e). The number of complexes observed on the slide as a function of time in the different conditions was variable from experiment to experiment, and thus, we refrain from making conclusions about the binding affinity of telomerase for its DNA substrate under the different conditions. Notably, we obtained similar results after collecting a much larger number of molecules ( ~ 10,000 to 20,000 molecules) from 20 fields of view per condition (Supplementary Fig. 4). While this collection method allows us to sample a larger number of molecules, the collection time and background correction are less precise (see "Methods"). Taken together, our smFRET experiments reveal that 6-thio-dGTP incorporation appears to induce telomerase stalling, thereby preventing telomerase from progressing through multiple rounds of repeat addition to achieve a stable low FRET state.

To further explore the possibility of 6-thio-dGTP incorporation inducing a bound non-productive telomerase complex, we next examined the dynamics of individual telomerase-DNA complexes during catalysis (Fig. 5). We analyzed ~100 single-molecule traces for each condition after 15 min incubation with either natural dNTPs or 6-thio-dGTP substitution or excess. The data were analyzed as individual FRET trajectories (Fig. 5a) and as a time-dependent heat map to capture the behavior of the population of complexes (Fig. 5b). As expected, the population without dNTPs showed a stable high FRET behavior before dropping to a zero FRET state caused by acceptor dye bleaching (Fig. 5a, b, top panels). Following 15 minutes of reaction with dNTPs, the complexes at an initial high FRET state of ~ 0.7 reduced to low FRET states of ~ 0.2–0.4 over time, consistent with histogram data (Fig. 5a, b, 2nd panels). In contrast, complexes supplied with 6-thio-dGTP instead of dGTP, or a 10-fold excess of 6-thio-dGTP over dGTP, largely persisted in the initial high FRET state and transiently sampled lower FRET

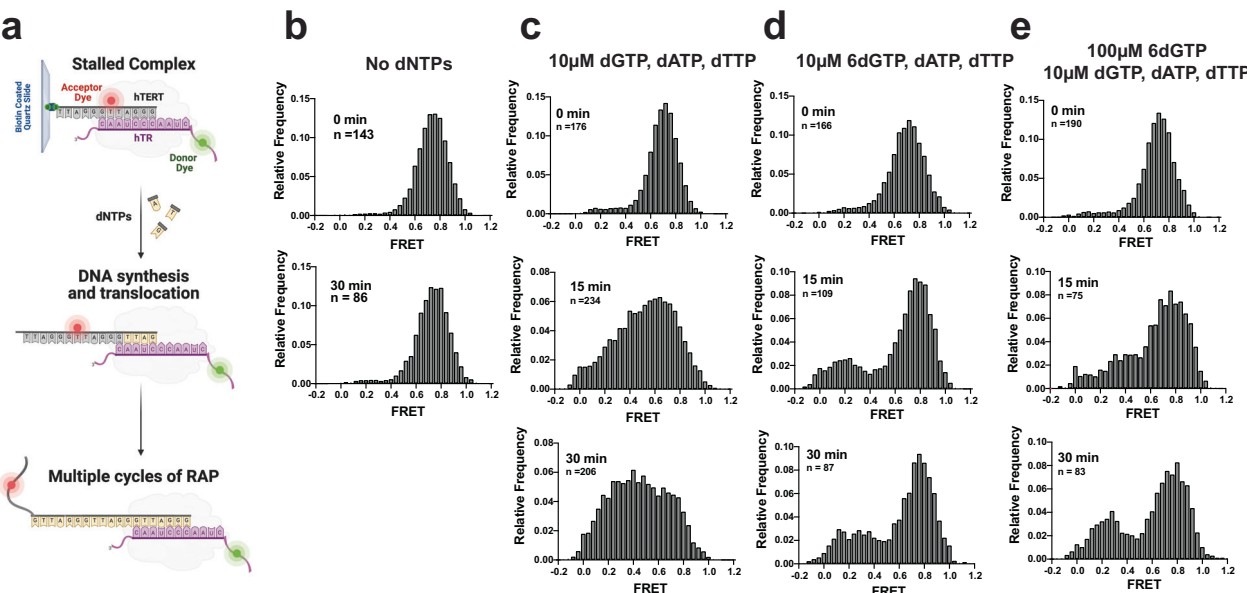

**Fig. 4 | Single molecule analysis reveals a non-productive telomerase complex after 6-thio-dG addition. a** Schematic of the experimental setup. LD555-labeled telomerase is pulled down on Cy5-labeled (TTAGGG)₃ substrate and immobilized on a quartz slide via a 5' biotin. The close proximity of the dyes permits FRET. Telomerase-catalyzed addition of dNTPs to the telomere substrate increases the distance between the dyes, causing a decrease in FRET dependent on the number of repeats added. Created in BioRender. Sanford, S. (2025) https://BioRender.com/wtax6ef. **b** FRET histograms of complexes prior to telomerase activity (top "stalled") and following a 30 min 'mock' activity incubation lacking dNTPs. **c** FRET

histograms of complexes incubated in the presence of dATP, dTTP, and dGTP collected prior to (top) and 15 and 30 min following dNTP addition. Shift from high to low FRET states reports on telomerase movement away from the acceptor dye during telomere elongation. **d** FRET histograms of complexes collected prior to (top) and 15 and 30 minutes after dATP, dTTP and 6-thio-dGTP (6dGTP) addition. **e** FRET histograms of complexes collected prior to (top) and 15 and 30 min after dATP, dTTP, dGTP and a 10-fold excess of 6-thio-dGTP addition. Source data are provided as a Source Data file.

states. Comparing the behavior of the telomerase-DNA complexes in the absence of dNTPs with complexes incubated with dATP, dTTP, and 6-thio-dGTP suggests that telomerase can incorporate the 6-thio-dGTP which in turn induces DNA dynamics (Fig. 5a, b compare top panel with 3rd and 4th panels). However, the inability of these complexes to exhibit RAP in the presence of 6-thio-dG prevents the accumulation of a low FRET signal that requires multiple rounds of telomere DNA repeat synthesis. In each condition, the sudden drop to zero FRET is a consequence of FRET dye photobleaching (Fig. 5a, black arrows). Our single-molecule FRET results support that telomerase can incorporate 6-thio-dGTP to complete the first cycle of telomere repeat synthesis, consistent with previous bulk DNA primer extension results[35]. This activity alters the dynamic properties of the telomere DNA within the telomerase complex without immediately promoting dissociation, but does not support the subsequent rounds of telomere repeat addition required to achieve the time-dependent stable low FRET state observed in the presence of normal dNTPs.

### Cellular 6-thio-dG treatment inhibits telomerase synthesis of telomeric DNA

Based on the strong biochemical and biophysical evidence that 6-thio-dGTP inhibits telomerase-mediated telomere extension, we next asked whether 6-thio-dG treatment inhibits telomerase activity in human cells. For this, we expressed a previously described hTR variant (C50/56A) harboring a 3'-AAAUCCAAAUC-5' template sequence so that telomerase adds 5'GTTTAG-3' repeats to telomeric ends instead of the wild-type (WT) 5'-GGTTAG-3' repeats. This allows us to distinguish telomerase synthesis of new repeats, or lack thereof, from the bulk telomeric DNA using probes specific for variant repeats (Fig. 6a)[51]. While this variant telomerase is less processive than wild-type, it is stimulated

by POT1 and TPP1[51] and inhibited by 6-thio-dGTP, similar to WT telomerase (Supplementary Figs. 5a–c). We transduced HCT116 cells with a lentivirus containing either an empty vector (EV) or a vector for C50/56A hTR expression, and 6 days post-transduction treated cells with increasing 6-thio-dG concentrations (0–5 μM) for 72 hr. Cells expressing C50/56A showed foci by fluorescent in situ hybridization (FISH) using PNA probes complementary to variant sequences, whereas EV controls did not (Fig. 6b and Supplementary Fig. 5d). 6-thio-dG treatment significantly reduced both the number of variant telomere foci per nucleus and variant foci sum intensity, indicative of the total amount of variant repeat DNA, compared to untreated cells in a dose-dependent manner (Fig. 6c, d). We also observed reduced wild-type TTAGGG sum intensity after 6-thio-dG treatment, likely due to inhibition of endogenous telomerase since these cells still express hTR (Supplementary Fig. 5e). As a control, we showed 6-thio-dG treatment did not alter staining intensity of centromeric PNA probes (Supplementary Fig. 5k). The decrease in telomeric synthesis could not be explained by cell growth reduction or senescence. We observed no significant differences in relative cell numbers between 0 and 1 μM 6-thio-dG, and between 2.5 and 5 μM 6-thio-dG, after 72 h treatments, yet variant telomere foci number and intensity were significantly reduced at each dose (Fig. 6c–e). Furthermore, the highest dose (5 μM) did not increase beta-galactosidase staining (b-gal) or nuclear area indicative of senescence and cell cycle exit (Fig. 6f, g). In agreement, a shorter 24 h treatment induced a smaller reduction in cell number, but still significantly reduced variant telomeric repeat foci (Supplementary Fig. 5f, g). While a 7-day 1 μM 6-thio-dG treatment reduced cell number, indicative of growth inhibition over time, the remaining cells showed decreased telomere variant foci and sum intensity, but no increase in nuclear area (Supplementary Fig. 5h–l). Finally, we show that WT or variant hTR expression does not

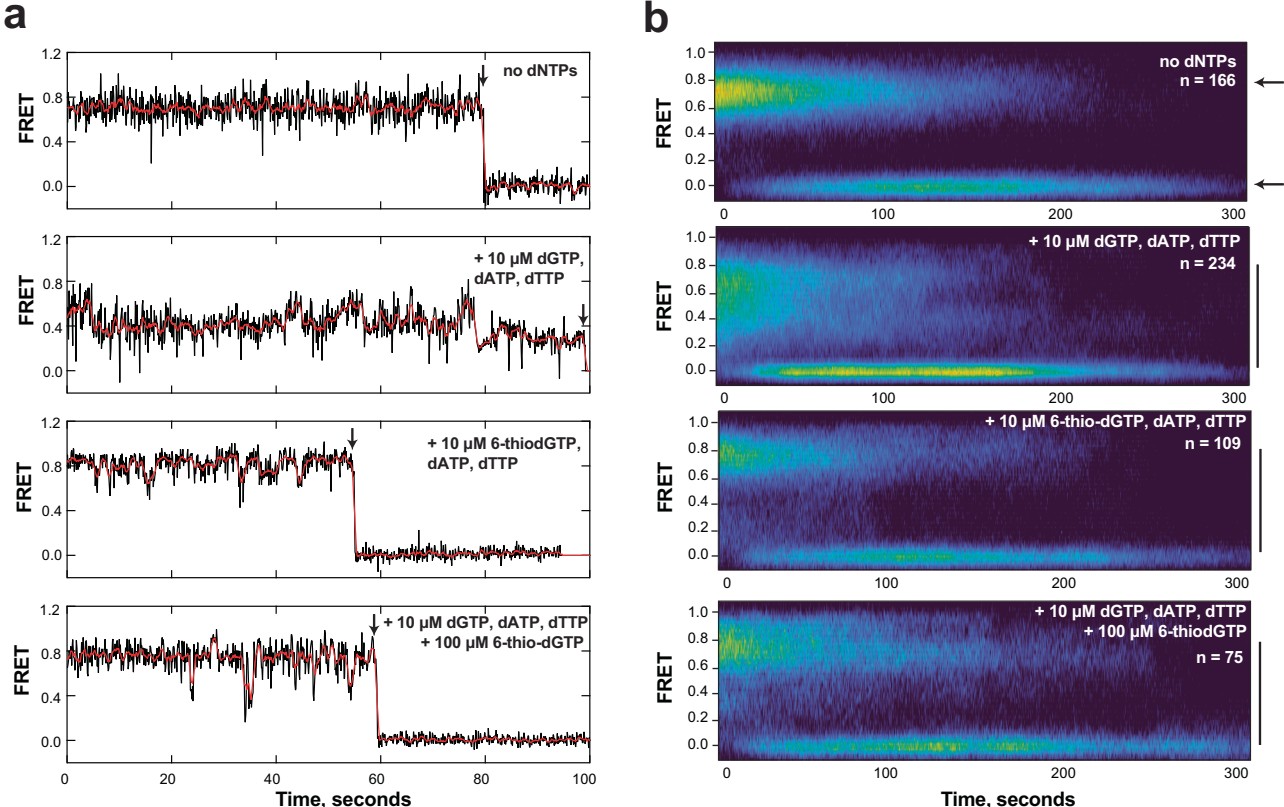

**Fig. 5 | Single molecule analysis reveals telomere DNA dynamics within non-productive telomerase complexes after 6-thio-dG addition. a** Representative traces of stalled single telomerase-DNA complexes (top), in the presence of dNTPs (second panel), with dATP, dTTP and 6-thio-dGTP (6dGTP) (third panel), and with dATP, dTTP, dGTP, and a 10-fold excess of 6-thio-dGTP (bottom panel). Data were collected 15 minutes after the addition of dNTPs. Raw data collected at 8.3 Hz framerate are shown in black, and a one-second moving average is overlaid in red. Black arrows indicate irreversible photobleaching of the FRET dyes. **b** Heat map analysis of the time-dependent FRET signal of ~100 individual telomerase-DNA complexes in each experimental condition. Small black arrows in the top panel indicate the FRET state of the stalled complex (upper arrow) and the FRET when the dyes have photobleached, also indicated by the signal at 0 FRET. In the presence of dNTPs (second panel), dATP, dTTP, 6-thio-dGTP (third panel) or dATP, dTTP, dGTP and 10-fold excess 6-thio-dGTP (bottom panel), broadening of the FRET distribution (solid black lines) indicates DNA dynamics within the telomerase-DNA complexes. Source data are provided as a Source Data file.

significantly alter sensitivity to 6-thio-dG (Supplementary Figs. 5m–o). Therefore, collectively our data reveal that 6-thio-dG reduction in new telomeric repeat synthesis is due telomerase inhibition and not cell senescence or reduced growth.

To further validate the ability of 6-thio-dG to inhibit telomerase activity in cells, we used a previously described assay of rapid telomerase-mediated telomere extension[52]. For this, 293T cells lacking hTR are transfected with expression vectors encoding both hTERT and hTR to generate super-telomerase cells, which show telomere elongation within 30 h, compared to controls transfected with vectors for expressing hTERT and eGFP (Supplementary Fig. 6). We observed that 6-thio-dG treatment prevented telomere elongation at 5 and 10 μM doses compared to untreated super-telomerase cells (Supplementary Fig. 6b, c). Lack of inhibition at 1 and 2.5 μM 6-thio-dG may be due to the shorter exposure time required for the super-telomerase assay (30 h) (Supplementary Fig. 6b), compared to the variant repeat addition assay (72 h) (Fig. 6c, d). Notably, 293T were more sensitive to 6-thio-dG treatment than HCT116 cells[52], which may be due to the very short telomeres in these cells (Supplementary Fig. 6a). Nevertheless, the super-telomerase assay confirms that 6-thio-dG inhibits new telomeric DNA synthesis by telomerase.

### Cancer cells with critically short telomeres are hypersensitive to 6-thio-dG

Based on our evidence that 6-thio-dG inhibits telomerase, we reasoned that telomerase-expressing cancer cells with shortened telomeres may be more sensitive to telomerase inhibition compared to cancer cells with long telomere reserves. We reported previously that telomerase inhibition due to 8-oxo-dGTP induces telomere losses and cell death in HeLa VST cells with very short telomeres (~3.7 kb), but not HeLa LT cells with long telomeres (~27 kb) after 3 days of treatment[53]. Therefore, we tested sensitivity to 6-thio-dG treatment of HeLa LT, HeLa VST, and the telomerase-negative cancer cell line U2OS, which uses the alternative lengthening of telomeres (ALT) pathway. HeLa VST cells showed the greatest reduction in colony formation after 9 days of treatment with 2.5 μM 6-thio-dG with a 2.3-fold reduction relative to untreated, compared to a 1.5-fold reduction for HeLa LT and a 1.7-fold reduction for U2OS (Fig. 7a). Interestingly, non-diseased retinal pigment epithelial (RPE) cells with and without hTERT expression exhibited similar 6-thio-dG sensitivity, and were less sensitive than the cancer cell lines (Supplementary Fig. 7a). These data suggest telomerase status alone does not determine sensitivity to 6-thio-dG and confirm a prior study showing cancer cell lines are more sensitive to 6-thio-dG compared to non-disease cell lines[26].

Next, we asked how 6-thio-dG impacts telomere integrity by analyzing telomeres on metaphase chromosomes following treatment. Previous studies demonstrated that 6-thio-dG treatment increases the appearance of dysfunctional telomeres marked by γH2AX colocalization with telomeric DNA in human cancer cell lines and mouse tumors[26], but the mechanism was unclear. The toxicity of 6-thio-dG treatment of HeLa VST cells limited the recovery of metaphase cells for chromosome analyses. Therefore, we treated HeLa LT and VST with a

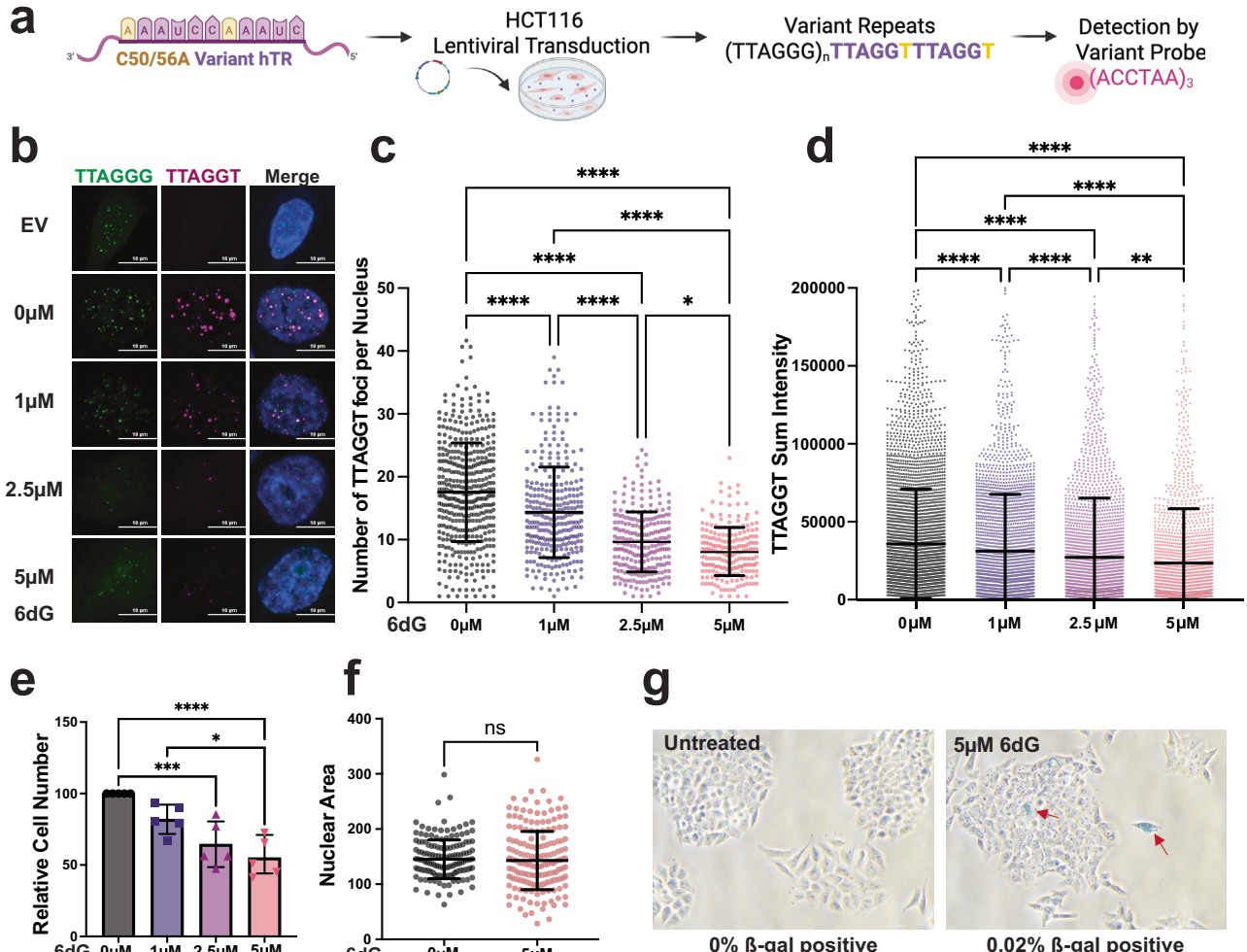

**Fig. 6 | 6-thio-dG suppresses telomerase elongation of telomeres in cells.**
**a** Schematic for expression of C50/56A hTR variant harboring the template sequence 3'-AAAUCCAAAUC-5' via lentiviral transduction into HCT116 cells, and detection of newly added (TTAGGT)n repeats with a 5'-Cy3-(ACCTAA)₃-3' probe by FISH. Created in BioRender. Sanford, S. (2025) https://BioRender.com/v6tf1o8.
**b** Representative images of interphase cells stained with PNA probes for wildtype and variant sequence 6 days post-transduction with empty vector (EV) or C50/56A hTR, and after 72 h treatment with 0, 1, 2.5 or 5 μM 6-thio-dG (6dG). **c** Quantification of the number of variant telomeric foci per nucleus from panel b. Error bars represent the mean ± s.d. from n cells analyzed as indicated by dots from 3 independent experiments. Statistical significance was determined by one-way ANOVA (*P = 0.0490; ****P < 0.0001). **d** Quantification of the variant telomere sum intensity. Error bars represent the mean ± s.d. of n variant telomere foci analyzed as indicated

by dots, from 3 independent experiments. Statistical significance was determined by one-way ANOVA (**P = 0.0026; ****P < 0.0001). **e** HCT116 cell counts obtained 72 hours after treatment with the indicated 6-thio-dG concentrations, relative to untreated cells. Error bars represent the mean ± s.d. from 4 independent experiments. Statistical significance was determined by one-way ANOVA (*P = 0.0103; ***P = 0.0010; ****P < 0.0001). **f** Size of nuclear area (μm²) obtained 72 hours after 0 or 5 μM 6-thio-dG treatment. Error bars represent the mean ± s.d. from the indicated n number of nuclei analyzed as indicated by dots, from two independent experiments. Statistical analysis by two-tailed t test (ns = not significant). **g** Representative images of β-galactosidase (β-gal) staining 72 hr after 0 or 5 μM 6-thio-dG treatment. Arrows indicate β-gal positive cells of 500 cells analyzed per condition from two independent experiments. Scale bar = 100 μm. Source data are provided as a Source Data file.

low dose 0.1 μM of 6-thio-dG over 20 days, which was generally well tolerated, allowing for recovery of metaphase chromosomes (Supplementary Fig. 7b). We reasoned 20 days should allow sufficient time to detect effects of telomerase inhibition on telomere maintenance. Chronic 6-thio-dG treatment significantly increased telomere losses, indicated by chromatid ends lacking telomere staining, only in HeLa VST cells, and increased telomere fragility in both cell lines (Fig. 7b–d). Fragile telomeres are marked by two or more telomeric foci at chromatid ends, indicative of impaired telomere replication, and correlate with telomere dysfunction[54]. 6-thio-dG treatment also increased the fraction of dysfunctional DDR+ telomeres in both cell lines (Fig. 7e, f). Finally, we conducted quantitative telomere FISH on interphase nuclei after chronic 6-thio-dG treatment to measure telomere length distribution. We confirmed previous reports that 6-thio-dG treatment induces telomere shortening in telomerase expressing cancer cells

(Fig. 7g)[26]. Collectively, these studies provide evidence that 6-thio-dG impairs telomere maintenance, leading to telomere shortening and telomere losses, especially in cancer cells with shortened telomeres. These studies also provide evidence that chronic 6-thio-dG treatment at low doses may impair telomeres replication, as evidenced by increased telomere fragility.

## Discussion
Numerous studies indicated that 6-thio-dG treatment causes telomere shortening and dysfunction in telomerase expressing cancer cells and tumors in vivo, however, the mechanism had been unclear[26,29,55]. Here, we provide direct evidence that 6-thio-dG treatment inhibits the synthesis of telomeric DNA by telomerase in cancer cells, and we uncovered the mechanism by which 6-thio-dGTP prevents telomerase from lengthening telomeres. Results from biochemical and

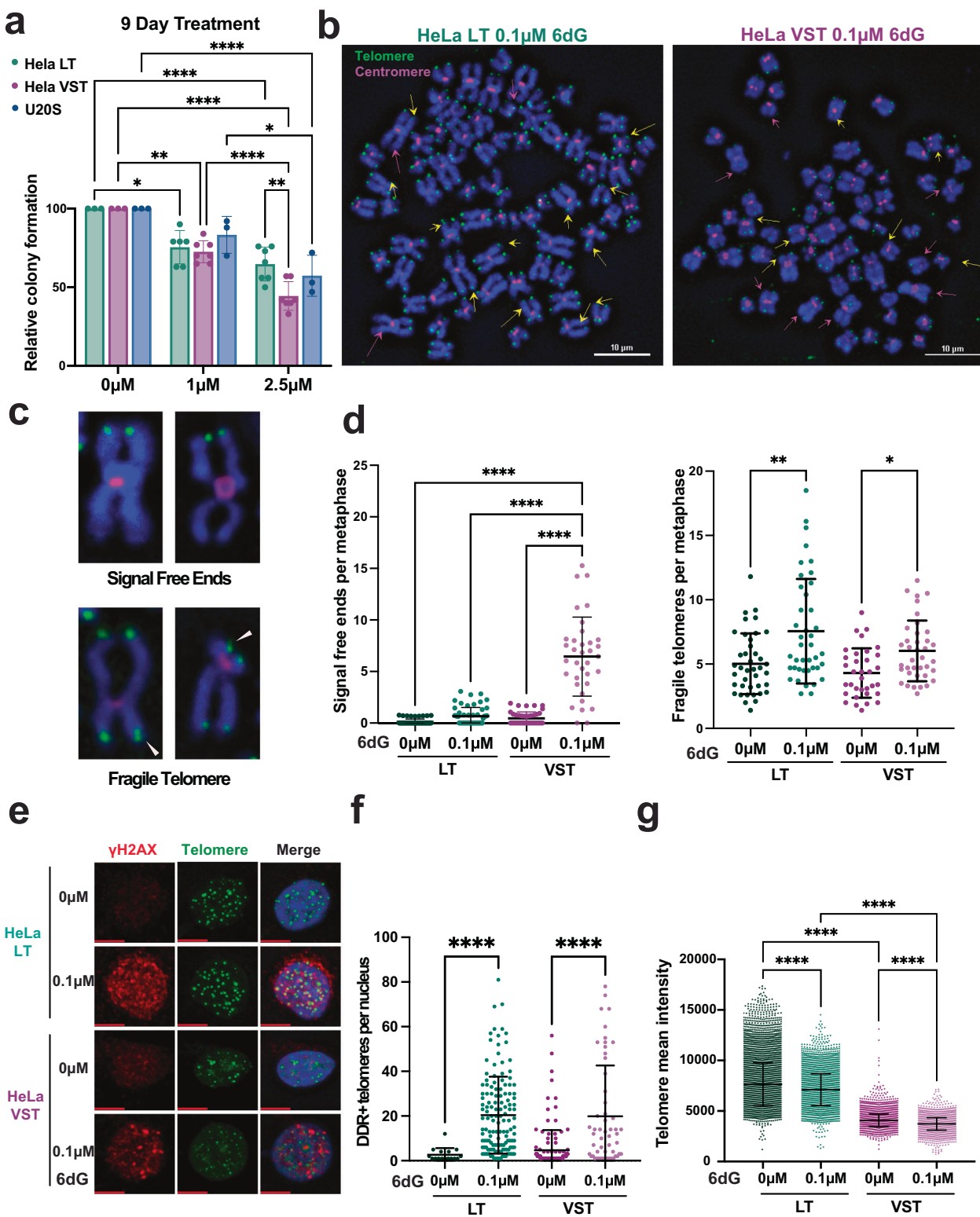

**a** 9 Day Treatment
- Hela LT
- Hela VST
- U2OS

**b** HeLa LT 0.1µM 6dG    HeLa VST 0.1µM 6dG
Telomere
Centromere

**c** Signal Free Ends

Fragile Telomere

**d**

**e** γH2AX   Telomere   Merge
HeLa LT 0µM / 0.1µM
HeLa VST 0µM / 0.1µM 6dG

**f**

**g**

biophysical assays are consistent with a mechanism in which the catalytic addition of 6-thio-dGTP causes telomerase to enter a non-productive "stalled" state without dissociating the enzyme from the telomeric DNA. We find that 6-thio-dGTP incorporation does not prevent telomerase binding to DNA or synthesis of the first repeat, but rather inhibits further repeat addition following enzyme translocation. Furthermore, we provide evidence that 6-thio-dGTP is likely a selective

telomerase inhibitor, since it does not significantly impair replicative DNA polymerase δ processivity. Our cellular studies reveal that 6-thio-dG treatment suppresses telomerase activity, promotes telomere shortening, and increases telomere losses, particularly in cancer cells possessing very short telomeres, consistent with failures in telomere restoration. Furthermore, cancer cells with shortened telomeres are more sensitive to 6-thio-dG, compared to cancer cells with long

**Fig. 7 | Telomerase expressing cancer cells with shortened telomeres are hypersensitive to 6-thio-dG. a** Colony formation efficiency following 9 days of treatment with 1 or 2.5 μM 6-thio-dG (6dG), relative to untreated for indicated cell lines. Error bars represent the mean ± s.d. from the number of independent experiments indicated by the dots. Statistical analysis by two-way ANOVA (HeLa LT 0 μM vs 1 μM *P = 0.0130, U2OS 1 μM vs 2.5 μM *P = 0.0288; HeLa VST 0 μM vs 1 μM **P = 0.0028; HeLa LT 2.5 μM vs VST 2.5 μM **P = 0.0049; ****P < 0.0001). **b** Representative images of FISH staining with WT telomere probes of metaphase chromosomes from indicated cell lines after 20 days of treatment with 0.1 μM 6-thio-dG. Images scored for telomeric signal-free ends (pink arrows) and fragile telomeres (yellow arrows). Green = telomeres and pink = centromeres. Scale bars, 10 μm. **c** Representative telomeric signal free ends and fragile telomeres (white arrowheads) from panel (**b**). **d** Quantification of telomeric signal-free ends and fragile telomeres per metaphase from panel (**b**). Error bars represent mean ± s.d. of

n metaphases, as indicated by dots, analyzed from three independent experiments. Statistical analysis by two-way ANOVA (*P = 0.0191; **P = 0.0015; ****P < 0.0001). **e** Representative images of γH2AX foci (red) and telomeric foci (green) stained by WT telomere FISH after 20 days of treatment with 0 or 0.1 μM 6-thio-dG. Colocalization shown as yellow foci. Scale bars, 10 μm. **f** Quantification average number of telomeres staining positive for γH2AX per cell after 20 days of treatment with 0 or 0.1 μM 6-thio-dG. Error bars represent the mean ± s.d. from n number of nuclei analyzed as indicated by dots from three independent experiments. Statistical analysis by one-way ANOVA (****P < 0.0001). **g** Quantification of WT telomere signal intensity from interphase cells after 20 days of treatment with 0 or 0.1 μM 6-thio-dG. Error bars represent the mean ± s.d. from n number of telomeric foci as indicated by dots from three independent experiments. Statistical analysis by two-way ANOVA (****P < 0.0001). Source data are provided as a Source Data file.

telomeres. Collectively, our studies provide evidence that 6-thio-dG impairs telomere maintenance in cancer cells by producing a non-productive, stalled telomerase complex after 6-thio-dGTP insertion.

How does 6-thio-dG prevent telomerase cycling required for adding multiple repeats? Translocation is a unique telomerase property and a vulnerable process that requires 1) release from the product DNA from the hTR template, 2) repositioning of the hTR template, and 3) re-priming to 4) synthesize another repeat[13] (Fig. 8a). We propose a working model in which 6-thio-dGTP insertion impairs the final step involving stable formation of a new DNA primer-RNA template hybrid, thereby producing a non-productive "stalled" complex. Several lines of evidence support this model. First, two independent assays confirm that 6-thio-dG in telomeric DNA fails to disrupt telomerase binding to the DNA. Second, our result that telomerase poorly extends substrates with a 3' terminal 6-thio-dG, but readily extends substrates with an internal 6-thio-dG outside the 4 bp RNA-DNA alignment region, is consistent with 6-thio-dG impairing proper re-annealing. Third, smFRET data reveals that telomerase exhibits dynamic movement of the DNA product in the presence of 6-thio-dGTP, but cannot achieve a stable low FRET indicative of multiple rounds of RAP. These data suggest that telomerase incorporates 6-thio-dGTP to complete a telomere DNA repeat, which then initiates dynamic DNA movements that must occur during translocation to reposition the DNA product with the RNA template. Fourth, smFRET data also show that telomerase remains engaged with the telomere following 6-thio-dGTP addition since we did not observe substantial loss of fluorescence signals indicative of enzyme dissociation. This finding is consistent with reports of DNA anchor sites within the telomerase catalytic core enzyme, which permit the 3'-end of the DNA to dynamically sample different conformations in the absence of enzyme dissociation[56]. Thus, even if 6-thio-dG alters the RNA-DNA hybrid integrity in the telomerase active site, a PLYQ motif in the telomerase TEN domain and amino acids in the IFD domain form an anchor that retains telomerase interactions with the DNA during translocation[56].

How might 6-thio-dGTP cause telomerase stalling? Modeling 6-thio-dG into the primer terminus of the telomerase CryoEM structure revealed no active site residues that directly contact the thio group, with the closest contact being R631 positioned well outside of hydrogen bonding distance at 5.1 Å away[36] (Fig. 8b–d). Similarly, modeling 6-thio-dGTP into the active site of telomerase as an incoming nucleotide shows no active site residues clashing with the modification, although R631 is ~ 3.0 Å away (Fig. 8e). Thus, available structural information suggests telomerase may be "blind" to modifications at the C6 of guanine due to minimal contacts with the major groove. However, NMR-based structural and thermal analyses of an 11-mer DNA duplex reveal that an internal 6-thio-dG-C base pair has slightly longer hydrogen bonds and an increased opening rate and instability compared to a duplex G-C[57]. Similarly, structures of DNA polymerase β with 6-thio-dGTP in the active site indicate a longer hydrogen bond distance between the thio group and the opposing dC (3.5 Å vs 3.0 Å for a

typical G-C pair), caused by base pair opening from the larger sulfur atom of the thio group[43]. The very short telomerase RNA-DNA hybrid may be vulnerable to the destabilizing effects of a 6-thio-dG-rC, especially when two consecutive 6-thio-dGs are inserted. We propose instability of a terminal 6-thio-dG-rC may increase fraying or opening and impair the ability of telomerase to extend synthesis following 6-thio-dG addition. In agreement, ~ 25% of the telomerase molecules terminate synthesis after inserting 6-thio-dGTP at that first rC of the hTR CCAAUC template[35], indicating some moderate impairment of extension from a 6-thio-dG-rC. However, after 6-thio-dGTP insertion, telomerase residues in the active site may help stabilize the hybrid, allowing for extension to the end of the template to complete the first repeat[56]. These stabilizing factors are lost once the enzyme translocates, thereby amplifying the 6-thio-dG destabilizing effects on new RNA-DNA hybrid formation for re-priming. This may explain why 6-thio-dGTP more strongly inhibits the telomerase repeat addition processivity needed for adding multiple repeats, compared to nucleotide addition processivity required to complete a single telomere repeat.

A previous model concluded telomerase 6-thio-dGTP insertion causes telomere dysfunction and shortening without inhibiting telomerase activity, based on results from the Telomerase Repeated Amplification Protocol (TRAP) with extracts from HCT116 cancer cells treated with 6-thio-dG thrice over a week[26]. The failure to detect decreased telomerase activity may reflect instability or loss of 6-thio-dGTP during extract preparation. Here, we report that direct measurement of telomerase activity in biochemical and biophysical assays after 6-thio-dGTP, or telomere repeat synthesis in intact cells after 6-thio-dG treatment, shows reduced telomerase activity. Our result that HeLa VST cells are more sensitive than HeLa LT cells to 6-thio-dG also supports telomerase inhibition, based on previous reports that cancer cells with short telomeres are more sensitive to telomerase inhibitors than cancer cells with long telomeres[53,58,59]. One proposed explanation is that cancer cells replicating with critically short telomeres may require telomerase extension at each cycle for viability[59]. In agreement, we find 6-thio-dG increases telomere losses in HeLa VST cells, but not HeLa LT cells. Invoking this model, we also propose non-diseased RPE-hTERT cells may lack these "critically short" telomeres, and therefore are not more sensitive than RPE cells to 6-thio-dG for 9 days, after which significant shortening due to cell division is not expected. Further evidence that 6-thio-dGTP inhibits telomerase activity derives from a study in NSCLC with activating EGFR mutations. EGFR tyrosine kinase inhibitor (TKI) osimertinib increased DDR+ telomeres and decreased telomerase activity and telomere lengths by reducing hTERT expression. EGFR-TKI resistant cells upregulate hTERT, and either hTERT depletion or 6-thio-dG treatment restores sensitivity[30]. Thus, the ability of 6-thio-dG to phenocopy hTERT loss in this setting is consistent with a mechanism of 6-thio-dGTP inhibiting telomerase activity versus telomerase insertion of 6-thio-dGTP causing telomere "uncapping"[60].

Other mechanisms of 6-thio-dG toxicity and telomere instability beyond telomerase inhibition are also possible. Our results with Pol δ

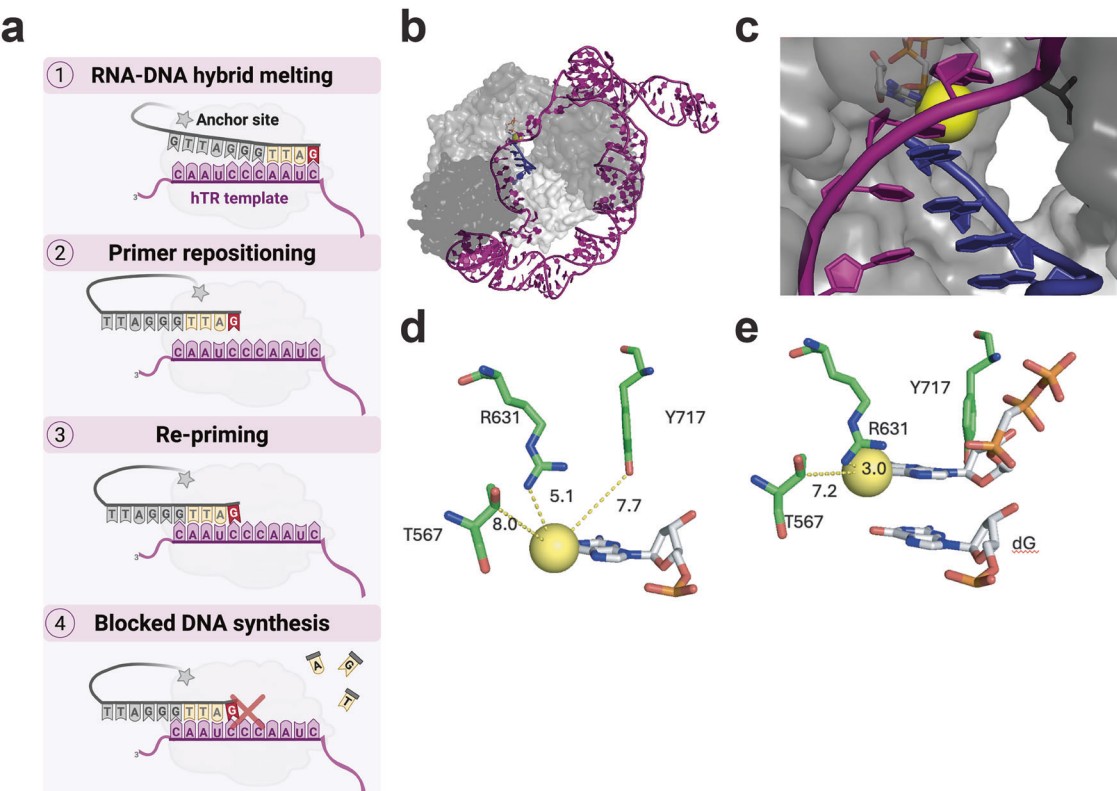

**Fig. 8 | Model for 6-thio-dGTP telomerase inhibition. a** Working model for how 6-thio-dGTP inhibits telomerase activity, see text for details. Telomerase hTERT in gray, anchor site depicted with the gray star, hTR template in purple, newly added dNTPs in yellow and 6-thio-dG in red. Created in BioRender. Sanford, S. (2025) https://BioRender.com/u2ldvlj. **b** Global structure of TERT (protein shown as a gray surface, hTR shown as purple sticks, telomere DNA shown in blue). **c** Closeup view of telomerase active site, showing cavity along the major groove side of the telomere. 6-thio-dGTP is modeled in for reference, with the thio group shown as a yellow sphere. Models in (**b** and **c**) were generated by aligning structures from human telomerase PDB code 7BG9 and Tribolium castaneum TERT with an incoming dGTP, PDB 6USR. **d** Model of 6-thio-dG at the primer terminus of a telomere (thio group shown as a yellow sphere, amino acids shown as green sticks, and DNA shown as white sticks). The closest active site contacts based on both structures are shown. The thio group is well accommodated by any protein residues. **e** Model of incoming 6-thio-dGTP, with closest contacts displayed. The thio group is ~3 angstrom away from R631, but it does not clash with its position and could potentially act as a stabilizing interaction. The model in (**d**) was generated from PDB code 7BG9, and the model in **e** was generated from overlaying PDB codes 7BG9 and 6USR, with measurements taken from the higher resolution tcTERT structures and residue numbering used from human TERT.

and PCNA agree with an early report that 6-thio-dGTP does not inhibit human DNA polymerases α and δ[61], suggesting they can incorporate 6-thio-dG at telomeres during DNA replication. However, we find 6-thio-dGTP insertion increased Pol δ stalling at two template positions, suggesting some ability to impair synthesis. Therefore, accumulated 6-thio-dG lesions may cause replication stress. The increased telomere fragility observed after chronic 6-thio-dG treatment is consistent with impaired telomere replication. Furthermore, 6-thioguanine residues in the genome can convert to S6-methylguanine via S-adenosylmethionine (SAM), which mispairs with T, a substrate for MMR[62]. Removal of T from the nascent strand allows for T reinsertion and futile cycles of repair. Acquired MMR deficiency is associated with thiopurine resistance in various cancers and relapse of acute lymphoblastic leukemia[61,63]. Therefore, processing by MMR may impair telomere replication and produce telomere fragility. However, because 6-thioG methylation is rare (~1 in $10^4$ 6-thioGs)[62], we argue this lesion likely does not contribute to acute 6-thio-dG toxicity, but may promote toxicity and telomere fragility after chronic exposures. Therefore, MMR and SAM status, in addition to telomerase status and telomere length, may influence the efficacy of 6-thio-dG and should be considered in future studies and evaluation of 6-thio-dG (THIO) use in the clinic.

Our studies have implications for mechanisms of thiopurine immunosuppression. Most prior studies focused on the ribonucleoside forms of 6-thioguanine and 6-mercaptopurine, since these are used to treat pediatric leukemias, chronic inflammatory disorders and for immunosuppression in organ transplants[23]. The main immunosuppressive mechanism is thought to be blockage of GTPase Rac1 signaling to suppress pro-inflammatory T cell response, partly by stimulating apoptosis[64]. GTPase inhibition may contribute to the greater toxicity of 6-thio-G used at the higher doses for cancer therapy, compared to 6-thio-dG[26]. However, 6-thio-G can be metabolized to 6-thio-dG and therefore, findings in our study are likely relevant to 6-thio-G toxicity. For example, our results raise the possibility that thioguanine immunosuppression may be partly due to 6-thio-dGTP inhibition of telomere maintenance in T cells. Interestingly, thiopurine treatment for inflammatory bowel disease is associated with an increased risk of acute myeloid leukemia (AML), myelodysplastic syndrome, and skin squamous cell carcinomas[23,64,65]. These same cancers are elevated in Short Telomere Syndromes caused by mutations in telomerase or related genes, and are proposed to involve impaired telomere maintenance in immune cells and other hematopoietic cell types[66]. Our results raise the possibility that thiopurine-induced telomerase inhibition in T cells may contribute to immunodeficiency.

In summary, we provide evidence that 6-thio-dG impairs telomere maintenance in cancer cells by inhibiting the ability of telomerase to re-engage the primer for additional repeat synthesis following 6-thio-dGTP insertion and telomerase translocation during cycling. Our study demonstrates how a greater mechanistic understanding of the unique properties of telomerase can be leveraged for developing selective

inhibitors like 6-thio-dG and potentially other nucleotide analogs for telomerase-targeted cancer therapy.

## Methods

This research and protocols used comply with the Institutional Biosafety Committee at the University of Pittsburgh.

### POT1 and TPP1 purification

Full-length human POT1 was expressed as a SUMOstar-(histidine)$_6$-POT1 fusion protein in baculovirus-infected SF9 cells (Thermo Fisher Scientific) as previously described[35]. POT1 was first purified using a His-Trap column (Cytiva) and eluted with 200 mM imidazole. After tag cleavage with SUMOstar protease (Ulp1 variant, LifeSensors), the protein was further purified by Superdex 200 size-exclusion chromatography (GE Healthcare) in 25 mM Tris-Cl (pH 8.0) and 150 mM NaCl buffer containing protease inhibitors (Roche Molecular Biochemicals). TPP1-N (amino acids 89–334) protein expression was induced with 0.8 mM IPTG in BL21(DE3) pLysS cells (Promega) for about 13 h at 24 °C, and then harvested by centrifugation at 4000 × g for 20 min. Cells were lysed in buffer (20 mM Tris pH 7.5, 500 mM NaCl, 10 mM imidazole) containing a protease inhibitor cocktail (Roche Molecular Biochemicals). After sonication, the lysate was centrifuged 40,000 × g for 75 min at 4 °C. The supernatant was filtered with a 0.2 micron filter and loaded onto a HisTrap FF column (GE LifeSciences), followed by washing and elution with 20 and 200 mM imidazole, respectively, using an ATKA Pure FPLC (Cytiva). TPP1-containing fractions were concentrated and exchanged into buffer (25 mM Tris pH 8.0, 150 mM NaCl, 5 mM DTT) with a Centricon-10 device (Amicon). Harvested protein was incubated with SUMO (Ulp1) protease (Invitrogen) to cleave the tag overnight at 4 °C with rotation at 20 rpm. Then, samples were loaded on a HiLoad 16/600 Superdex 200 column (Cytiva), and eluted TPP1-containing fractions were collected and pooled. Protein concentrations were determined by Bradford Assay (BioRad), and purity was determined by SDS–PAGE and Coomassie staining.

### Telomerase and Halo-tagged telomerase preparation

Immuno-purification of telomerase was as described[35]. HEK-293T (ATCC) cells were grown to 90% confluency in Dulbecco's modified Eagle's medium (Gibco) supplemented with 10% FBS (Hyclone) and 1% penicillin-streptomycin (Corning) at 37 °C and 5% CO$_2$. Cells were transfected with 10 µg pSUPER-hTR plasmid (or with C50/56A mutations) and 2.5 µg pVan107 hTERT or pVAN107 Halo-3xFLAG-hTERT plasmid diluted in 625 µl Opti-MEM (Gibco) mixed with 25 µl of Lipofectamine 2000 (ThermoFisher) diluted in 625 µl of Opti-MEM. Cells expressing hTR and 3xFLAG-tagged human hTERT were harvested 48 h after transfection, trypsinized and washed with PBS, and then lysed in CHAPS buffer (10 mM Tris-HCl, 1 mM MgCl$_2$, 1 mM EDTA, 0.5% CHAPS, 10% glycerol, 5 mM β-mercaptoethanol, 120 U RNasin Plus (Promega), 1 µg/ml each of pepstatin, aprotinin, leupeptin and chymostatin, and 1 mM AEBSF) for 30 min at 4 °C. 80 µl of anti-FLAG M2 bead slurry (Sigma) (per T75 flask) was washed 3X with 10 volumes of human telomerase buffer (50 mM Tris-HCl, pH 8.0, 50 mM KCl, 1 mM MgCl2, 1 mM spermidine and 5 mM β-mercaptoethanol) in 30% glycerol and harvested by centrifugation for 1 min at 1370 × g and 4 °C. The bead slurry was added to the cell lysate and nutated for 4-6 hours at 4 °C. The beads were harvested by 1 min centrifugation at 1370 × g, and washed 3X with human telomerase buffer with 30% glycerol. For the Halo-FLAG-telomerase, after the washes, 1 µM Janelia Fluor® HaloTag® Ligand 635 was added to the slurry for 1 hour, nutated at 4 °C, and then washed 3X more. Telomerase was eluted from the beads with 2X the bead volume of 250 µg/mL 3X FLAG® peptide (Sigma Aldrich) in telomerase buffer containing 150 mM KCl. The bead slurry was nutated for 30 min at 4 °C. The eluted telomerase was collected using Mini Bio-Spin® Chromatography columns (Bio-Rad). Samples were flash-frozen and stored at −80 °C.

### Dot blot quantification of telomerase concentration

The concentration of telomerase pseudoknot RNA in the eluted telomerase preparations was measured as described previously[35]. Briefly, a serial dilution of in vitro transcribed pseudoknot region of hTR (Supplementary Fig. S1) was prepared as standards for quantification (0.1, 0.5, 1, 5, 10, 50, 100, 250 fmol/µl). An aliquot of each standard and telomerase (10 µl) was added to 90 µl of formamide buffer (90% formamide, 1 × Tris–borate EDTA (TBE)). Samples were incubated at 70 °C for 10 min and then placed on ice. Positively charged Hybond H + membranes and Whatman filter papers (GE Healthcare Life Sciences) pre-incubated with 1 × TBE were assembled onto the GE manifold dot blot apparatus, and samples were loaded onto the membrane via vacuum blotting. The membrane was air dried and then UV-crosslinked using a Stratagene Stratalinker 1800 with the Auto-Crosslink program. The membrane was prehybridized at 55 °C in 25 ml of Church buffer (1% BSA, 1 mM EDTA, pH 7.5, 500 mM Na$_2$HPO$_4$ pH 7.2, 7% SDS) for 30 min. A total of $1 \times 10^6$ CPM of $^{32}$P-labeled hTR oligonucleotide probe (Supplementary Table S1) was added to the hybridization buffer and incubated overnight at 55 °C. The membrane was washed 3 × with 0.1 × SSC, 0.1 × SDS buffer. After vacuum sealing, the membrane was exposed to a phosphorimager screen for 1–3 h and imaged using a Typhoon scanner (Cytiva). ImageQuant TL 8.2 was used to quantify the blot intensities for the standard curve.

### End-labeling of DNA primers for telomerase assay

50 pmol of PAGE-purified DNA oligonucleotides (IDT) (Supplementary Table 1, primers 1-7 and A5) were labeled with γ$^{32}$P ATP (Perkin Elmer) using T4 polynucleotide kinase (NEB) in PNK Buffer (70 mM Tris-HCl, pH 7.6, 10 mM MgCl$_2$, 5 mM DTT) in a 20 µl reaction volume. The reaction was incubated for 1 h at 37 °C followed by heat inactivation at 65 °C for 20 min. Unincorporated γ$^{32}$P ATP was removed with G-25 spin columns (GE Healthcare).

### Telomerase activity assay

Reactions (20 µl) contained 1X human telomerase buffer, 5 nM of $^{32}$P-end-labeled primer as indicated in the Figure legends and dNTPs at cellular relevant concentrations (24 µM dATP, 29 µM dCTP, 37 µM dTTP, 5.2 µM dGTP). Reactions in Fig. 1 with primer A5 included increasing concentrations of 6-thio-dGTP (TriLink) as indicated. The reactions were started by adding 3 µl of immunopurified telomerase, incubated at 37 °C for 1 h, then terminated with 2 µl of 0.5 mM EDTA and heat-inactivated at 65 °C for 20 min. An equal volume of loading buffer (94% formamide, 0.1× TBE, 0.1% bromophenol blue, 0.1% xylene cyanol) was added to the reaction. Samples were heat denatured for 10 min at 100 °C and loaded onto a 14% denaturing acrylamide gel (7 M urea, 1× TBE) and electrophoresed for 90 min at constant 38 W. Samples were imaged using a Typhoon phosphorimager (Cytiva). Percent primer extension was calculated with ImageQuant TL 8.2 by measuring the intensity of each product band and dividing by the total radioactivity in the lane or total products, as indicated in the figure legends. Processivity was calculated as previously described[35].

### Recombinant human proteins and substrates for DNA Pol δ primer extension

Recombinant human RPA, PCNA, RFC, and pol δ holoenzyme were obtained as previously described[67–69]. The concentration of active RPA was determined as described previously[68]. Oligonucleotides were synthesized by Integrated DNA Technologies (Coralville, IA) and purified on denaturing polyacrylamide gels. The concentrations of DNAs were determined from the absorbance at 260 nm using the calculated extinction coefficients. For annealing, the template and complimentary primer strands were mixed in equimolar amounts in 1X Annealing Buffer (10 mM Tris-HCl, pH 8.0, 100 mM NaCl, 1 mM EDTA), heated to 95 °C for 5 min, and allowed to slowly cool to room temperature.

## DNA Pol δ primer extension assays

All primer extension experiments were performed at 25 °C in 1x Replication Buffer (25 mM HEPES, pH 7.5, 10 mM Mg(OAc)$_2$, 125 mM KOAc) supplemented with 1 mM DTT and 1 mM ATP. For all experiments, the final ionic strength was adjusted to 200 mM by adding an appropriate KOAc amounts. Cy5-labeled P/T DNA substrate (250 nM) (Supplementary Fig. 2a) was preincubated with 1 μM neutravidin, followed by RPA addition and equilibration for 5 min. Next, PCNA (250 nM homotrimer), ATP (1 mM), and RFC (250 nM heteropentamer) were added and incubated for 5 min. Finally, dNTPs (46 μM dATP, 9.7 μM dGTP, 48 μM dCTP, 67 μM dTTP) were added, followed by Pol δ addition (8.80 nM heterotetramer) to initiate DNA synthesis. We substituted dGTP (9.7 μM) with 6-thio-dGTP (9.7 μM) where indicated in the Figure legends. Reaction aliquots were removed at various time points and quenched with 62.5 mM EDTA, pH 8.0, 2 M Urea, 50% formamide supplemented with 0.01% (wt/vol) tracking dyes. Primer extension products were resolved on 16% polyacrylamide sequencing gels. Samples were heated at 95 °C for 5 min and iced for 5 min before loading on the gel. The gels were imaged with a Typhoon Model 9410 imager. The fluorescence intensity in each DNA band was quantified with ImageQuant (Cytiva), and divided by the sum of the intensities for all of bands in the lane to calculate percent primer extension. The probability of insertion, $P_i$, for each dNTP insertion step, $i$, after $i = 1$ was calculated as described previously[70,71].

## Electrophoretic mobility shift assay

Increasing concentrations (0 – 20 nM) of Halo-FLAG-telomerase conjugated to Janelia fluorophore (JF-635) were mixed with 2.5 nM flourescin labeled telomere substrates containing a 3′ terminal dG or 6-thio-dG (EMSA oligos, Supplementary Table 1) in binding buffer (20 mM HEPES-KOH pH 7.5, 100 mM KCl, 0.5 mM MgCl2, 0.5 mM TCEP-HCl, 0.05% v/v IGEPAL Co-630, 8% glycerol, 50 μg/mL bovine serum albumin) and incubated for 30 min at room temperature (22 °C). The reactions (10 μl) were loaded a 4–20% polyacrylamide-TBE gel in 1x TBE buffer and electrophoresed at 200 V. Gels were imaged using the near-IR setting of a Typhoon scanner (GE). Bands within the gel showing co-localized fluorescein labeled DNA with JF-635 labeled Halo-telomerase were considered bound substrate, and bands within the gel showing non-colocalized fluorescein DNA were considered unbound substrate. Halo-telomerase that failed to migrate out of the wells was excluded from analysis since this may represent improperly folded telomerase or hTERT lacking hTR. Bound and unbound susbtrates were quantified with ImageQuant, and % bound was calculated as bound substrate divided by total substrate.

## C-Trap confocal imaging and processing

A Lumicks C-Trap optical tweezers/three color confocal microscope was used to measure telomerase binding to telomeric substrates at the single molecule level. Lambda DNA was used as a scaffold tightrope between two biotinylated beads as described[47]. The telomeric single stranded 5′-(GGTTAG)$_3$-3′ or scramble control overhang substrate was prepared by ligating a telomeric Fluorescein-labeled oligonucleotide with a terminal 3′G or 6-thio-dG or non-telomeric oligonucleotide (see Supplementary Table 1, EMSA oligos) to a 13-nucleotide gap complementary to the lambda DNA scaffold generated by nickase digestion as described previously[47]. Briefly, 4.4-4.8 μm polystyrene beads were immobilized in optical traps, moved to a channel in the flow cell containing DNA to generate a single DNA tether between them, and then moved to a channel with Halo-FLAG-tagged telomerase conjugated with Janelia fluorophore (JF-635) to measure binding kinetics to the telomeric sequence (workflow previously described[72]). The Fluorescein label on the telomeric DNA served as a fiducial marker to identify binding events at the position of telomere sequences. Fluorescein was excited with a 488 nm laser and emission collected with a

500–550 nm band pass filter, and HaloTag-JF-635 was excited with a 638 nm laser and emission collected with a 650– 750 nm band pass filter. The laser power was set to 5% in each case, corresponding to 1–1.5 μW at the objective, and collected at 0.1 millisecond exposure per 100 nm pixel in kymograph mode. All data were collected with a 1.2 NA 60x water immersion objective and photons measured with single-photon avalanche photodiode detectors. Kymograph processing was performed in custom Lakeview software from Lumicks, where the binding time and dissociation times of on-target binding events were measured, sorted by duration, and converted to a cumulative residence time distribution (CRTD) for modeling in Graphpad Prism. In these CRTD plots, the Y values represent the fraction of events remaining after the indicated time on the X-axis, and the off rates and lifetimes were quantified in this analysis by the double-exponential function: $Y = \text{Plateau} + \text{SpanFast}*e(-\text{KFast}*X) + \text{SpanSlow}*\exp(-\text{KSlow}*X)$ where plateau represents the minimum X value, SpanFast/SpanSlow represent the percentage contribution of each rate towards the fit, and Kfast/Kslow representing the off rates observed. For fair comparison of the conditions, the ratio of slow:fast off rates was each constrained to ~50%. Weighted average lifetimes reported are generated by multiplying the percentages of each lifetime by their durations and then summing the values.

## LD555-labeled telomerase reconstitution and Cy5 DNA primer labeling

Human telomerase LD555 labeled at U42 on hTR was produced as described previously[49,50]. Briefly, telomerase was reconstituted in rabbit reticulocyte lysate supplied with in vitro transcribed CR4/5 239-329 and U42 LD555 (Lumidyne) PK 32-195 fragments as well as pNFLAG hTERT plasmid and purified via an N-terminal FLAG tag immunopurification. Single-stranded primer (TTAGGG)$_3$ was purchased with a 5′ biotin and a C6 amino modifier at T6 from the 3′ end (IDT Inc.) (Supplementary Table 1, smFRET primer). The oligonucleotide was labeled with Cy5-NHS (Cytiva) and HPLC purified using a C8 column.

## Single-Molecule FRET and data acquisition and processing

**Slide preparation.** Slides were prepared and assembled using biocompatible double-sided tape as previously described[40,49]. Quartz slides (Finkenbeiner Inc.) were cleaned in boiling water and with Alconox detergent. Then, slides were sonicated for 20 min in glass holders containing 10% (w/v) Alconox, rinsed with milliQ water and sonicated for 5 min in ultrapure water. Next, slides were sonicated for 15 min in acetone, transferred into 1 M KOH and sonicated for 20 min. Piranha solution was prepared by mixing 450 mL concentrated Sulfuric Acid and 150 mL (3:1 ratio) of 30% H$_2$O$_2$. Slides were treated with Piranha solution for 20 minutes, rinsed, and placed in methanol. Silanization of slides was by adding silanization solution (mixture of 100 ml methanol, 5 ml glacial acetic acid and 3 ml of (2-aminoethyl)-3-aminopropyltrimethoxysilane (UCT)) and sonication for 1 min, then incubation for 20 min at room temperature. For slide pegylation, 400 mg of mPEG-Succinimidyl Valerate MW 5000 (Laysan Bio, Inc.) were dissolved in 800 μl of 0.2 μM filtered 0.1 M sodium bicarbonate, and 2 mg of Biotin-PEG-Succinimidyl Valerate MW 5000 (Laysan Bio, Inc.) were dissolved in 200 μl filtered 0.1 M sodium bicarbonate. The mPEG and Biotin-PEG solutions were combined (PEG solution) dissolved by 30 s vortexing and ultracentrifuged at 13,000 x g for one minute. After silanization, the slides were rinsed with methanol, dried with 0.2 μM filtered ultrapure nitrogen gas and placed in a humidor box. 150 μl of PEG solution were applied to each slide and covered with a coverslip, and the slides were incubated overnight in the dark. The coverslip and excess PEG solution were then rinsed off with milliQ water, and the slides were dried using nitrogen gas. Sample channels were assembled on the pegylated slide surface using biocompatible tape strips as spacers. Plasma-

treated coverslips were placed to cover the tape strips and channels as the upper channel face.

**smFRET Telomerase reactions.** For telomerase reactions, (5 µl) U42-LD555 labeled telomerase (3 µl) were mixed with a 1 nM biotinylated Cy5-labeled smFRET primer (1 µl of 5 nM stock) (Supplementary Table 1) in T50 buffer (50 mM Tris-HCl pH 8.3, 50 mM LiCl) and 1 µl Telomerase Imaging Buffer in LiCl (TIBL) (50 mM Tris-HCl pH 8.3, 50 mM LiCl, 10 mM Trolox, 3 mM $MgCl_2$, 0.8% Glucose). Reactions were incubated at room temperature for 30 min, diluted to 40 µl with TIBL and immobilized on a PEGylated quartz slide. The solution was incubated on the slide for 10 minutes before washing with 60 µl of TIB KCl (TIBK) (50 mM Tris-HCl pH 8.3, 50 mM KCl, 10 mM Trolox, 3 mM $MgCl_2$, 0.8% Glucose) supplemented 1% (v/v) of freshly prepared 'Gloxy' solution (200 µg/ml catalase and 100 mg/ml glucose oxidase in T50 buffer at 100% stock concentration). After the Cy5-labeled DNA primers complexed with telomerase were immobilized on the microscope slides and 'stalled' complex data were collected, channels were washed 3x with 20 µl activity buffer containing 10 µM dATP, 10 µM dTTP, and either 10 µM dGTP replaced with 10 µM 6-thio-dGTP or excess 100 µM 6-thio-dGTP added in TIBK, as indicated.

**Data acquisition and analysis.** Imaging fields containing several hundred molecules were collected on a home-built prism-type Total Internal Reflection Fluorescence (TIRF) microscope equipped with an Andor Ixon EMCCD camera at 8.3 frames/second for smFRET experiments. Experiments used custom Labview software to acquire data and control laser shutters. Experimental data were collected in two different ways. As a first approach, twenty unique movies (~2 s) were collected in different fields of view for each time point (5, 15, 30 min reactions) and condition (Supplementary Fig. 4) and in a second approach, long movies (300 seconds) were collected for real-time smFRET trace analysis (Figs. 4 and 5). In both cases, raw single-molecule video data was analyzed using custom Matlab scripts to extract background corrected donor and acceptor dye intensity values for each molecule. For long smFRET movies, each individual trajectory was further corrected for background and signal crosstalk and only FRET traces that displayed single step photobleaching for both donor and acceptor were included for downstream analysis. FRET ratios were calculated using the equation $FRET = I_A/(I_A + I_D)$ where $I_A$ is the background corrected acceptor intensity and $I_D$ is the background corrected donor intensity. Histograms and trace representations were generated using custom written MATLAB scripts and GraphPad Prism. All smFRET histograms were generated by sampling the same 2-second window from each individual telomerase-DNA complex. Telomerase complexes that are labeled with LD555 (donor dye) and stick non-specifically to the surface generate an artifactual zero FRET peak that was removed from the histograms for clarity. Zero-peak removal was performed by fitting the zero FRET peak in each histogram with a single Gaussian function in GraphPad Prism, fixing the center and width parameters but floating the amplitude. The fit parameters were then used to subtract this peak from the FRET histograms shown in Fig. 4. For smFRET trace analysis, individual single-molecule traces were pre-analyzed using custom written Matlab software to perform fine correction of dye-cross talk and background correction. FRET traces were plotted in GraphPad Prism with a 1-second moving average overlay to guide the eye. FRET trace heat maps were generated using a custom Matlab script that bins each FRET value extracted for each frame across the ~100 telomerase-DNA complexes.

**Cell Culture.** RPE1-hTERT cells were cultured in Dulbecco's modified Eagle medium DMEM/F12 (Gibco) with 10% FBS (Gibco) and 1% penicillin/streptomycin. HeLa LT, HeLa VST, U2OS, HCT-116, HEK293-T, HEK293-FT cells were cultured in DMEM supplemented with 10% fetal

bovine serum (Gibco), 50 units/ml penicillin, and 50 units/ml streptomycin (Gibco) at 37 °C in humidified chambers with 5% $CO_2$ and 5% $O_2$, except for HEK293T and HEK293-FT cells which were maintained at 20% $O_2$. All cell lines were from ATCC, except HeLa LT and HeLa VST were a generous gift from Dr. Roderick O'Sullivan (University of Pittsburgh), and HEK293-FT were from ThermoFisher. Cell treatments with 6-thio-dG were as indicated in the figure legends.

**Generation of WT hTR and C50/56A variant hTR expressing cell lines.** HCT116 cell lines expressing WT or C50/56A hTR were prepared as previously described[51]. Vectors pLV-IU1-hTR-CMV-GFP and pLV-IU1-C50/56AhTR-CMV-GFP were generated with expression cassettes for Puromycin selection as described[51]. The empty vector (EV) lacks the hTR sequence. Lentivirus was packaged in HEK293FT cells using the vector plasmids, the Delta 8.9 packaging plasmid, and the VSV.G envelope plasmid as described[73]. The media from HEK293-FT cells was replaced with DMEM containing 1% BSA without antibiotics, and the cells were transfected with the plasmids and polyethylenimine (PEI) at a ratio of 2 µg:1 µg in Optimem. Supernatant was collected at 48 and 72 h, filtered using a 0.45 µm sterile filter (Thermo Fisher), and concentrated using Amicon Ultra Centrifugal Filters (15 mL, Millipore Sigma). Lentivirus was stored at − 80 °C. HCT116 cells were transduced with the WT hTR, C50/56A hTR, or EV lentivirus co-expressing GFP at an MOI of 5 in the presence of 8 µg/mL polybrene. Media was replaced with the standard media at 24 h post-transduction. At day 2 post-transduction, cells were selected with 2 µg/ml puromycin. On day 3 post-transduction, cells were treated for 72 hours with DMSO or 6-thio-dG at concentrations indicated in the figure legends.

**PNA probes.** All PNA probes were synthesized by PNABio. The WT telomere probe was Alexa-488(TelC-Cy3, F1004) containing 3 repeats of 5'-CCCTAA-3'. The pan-centromere probe was Cy5 conjugated (CENPB-488, F3005) 5'-ATTCGTTGGAAACGGGA-3'. The variant telomere probe was custom made and conjugated to Alexa 647 5'-ACCTAAACCTAAACCTAA-3' as described previously[51].

**Growth analyses.** For cell counting experiments, cells were plated at a low density in six-well plates with 6-thio-dG as indicated and recovered for the indicated amount of time (24 hr, 72 hr, or 7 days). Cells were detached from the plates, resuspended and counted on a Beckman Colter Counter or a Denovix CellDrop FLi. Each experiment included three to four technical replicates, which were averaged.

**Senescence-associated β-galactosidase assay.** β-galactosidase activity was measured according to the manufacturer's instructions (Cell Signaling). Briefly, cells were washed with PBS, and then fixed at room temperature for 10 min. After two rinses with PBS, cells were incubated overnight at 37 °C with X-gal staining solution and no $CO_2$. Images were acquired with a Nikon brightfield microscope with a DS-Fi3 camera. Images were scored in NIS-Elements (Nikon). At least 500 cells were counted per condition for each experiment.

**Colony formation assays.** Cells were plated in 6-well plates and treated with varying doses of 6-thio-dG as indicated in the figure legends. After 9 days, the colonies were fixed on ice in 100% methanol, stained with crystal violet solution and then counted manually.

**Super-telomerase telomere lengthening assay.** Transfections were conducted Lipofectamine LTX with Plus Reagent (Invitrogen) in Opti-MEM (Gibco) according to the manufacturers' instructions. The day before transfection, 2 million TERC-null 293 T cells described previously[52] were seeded per well of a six-well dish, then transfected with 2.5 µg total DNA per well with pmax eGFP control vector (Lonza), pBS U3-TERC-500 (Addgene 28170) to overexpress TERC and/or

PCDNA3.1-3XHA-TERT (Addgene 51637) to overexpress TERT as indicated in the figure legend. For conditions under which TERT was transfected in addition to TERC/eGFP, the total DNA input was kept the same, and a 5:1 mass ratio of TERC to TERT vector was used. 18 h post-transfected cells were treated with DMSO or concentrations of 6-thio-dG indicated in the figure legend for 30 h before harvesting. Super-telomerase extracts were prepared 2 days post-transfection. Genomic DNA was isolated using the PureLink Genomic DNA Mini Kit (Invitrogen). Briefly, 1–3 µg DNA was digested with RsaI (NEB) and HinfI (NEB) for 2–3 h at 37 °C and loaded onto a 0.6% agarose gel, followed by Southern blotting onto Hybond-N+ membrane (Amersham). Detection was conducted using the TeloTAGGG Telomere Length Assay Kit (Roche) with either the telomeric probe provided in the kit. After hybridization, detection was performed as according to the manufacturer's protocol (Roche). Quantification was performed using ImageJ by measuring the ratio of high molecular weight signal to low molecular weight signal as described previously[52]. This ratio was normalized so that the eGFP/TERT expressing samples were set to 0 and the untreated TERC/TERT expressing samples were set to 1. Statistical analysis was performed using GraphPad Prism version 9.1.0.

**Immunofluorescence and fluorescence in situ hybridization.** Cells were seeded on coverslips in 6-well plates and treated with DMSO or 6-thio-dG as indicated in the figure legends. Following recovery cells were washed twice with PBS and fixed with 4% formaldehyde in PBS for 10 min at room temperature. Cells were then rinsed with 1% bovine serum albumin (BSA) in PBS and washed twice with PBS. Cells were then permeabilized with PBS-Triton 0.2% (3 × 5 min washes) before blocking for 1 h at room temperature with 10% normal goat serum, 1% BSA, and 0.1% Triton X. The indicated primary antibodies were incubated overnight at 4 °C. The next day, coverslips were washed with PBS-Triton 0.2% and incubated with the respective secondary antibodies, followed by 3x washes with PBS-T. For fluorescence in situ hybridization (FISH), the cells were refixed with 4% formaldehyde, rinsed with 1% BSA in PBS and then dehydrated with 70%, 90% and 100% ethanol for 5 min. Telomeric PNA probe was diluted 1:100 (PNABio) prepared in 70% formamide, 10 mM Tris-HCl pH 7.5, 1x Maleic Acid buffer, 1x $MgCl_2$ buffer and boiled for 5 min before returning to ice. Coverslips were hybridized with the probe first by heating at 75 °C to denature the DNA, followed by hybridization in humid chambers overnight at 4 °C. The cells were washed twice with 70% formamide and 10 mM Tris-HCl pH 7.5, three times with PBS-T and mounted with antifade gold containing DAPI. Image acquisition was performed with a Nikon Ti inverted fluorescence microscope. Z stacks of 0.2 µm thickness were captured, and images were deconvolved using the NIS Elements Advance Research software algorithm.

**Metaphase spreads and quantitative telomere FISH.** Chromosome metaphase preparations and quantitative telomere FISH were performed as described previously. Cells were treated with 0.05 µg/mL colcemid for 2 h and harvested with trypsin, followed by incubation in 75 mM KCl hypotonic solution for 8 min at 37 °C and fixation in methanol and glacial acid acetic (3:1). Cells were then dropped on slides pre-treated with methanol. After drying overnight, cells were fixed with 4% PFA for 10 min at room temperature, followed by treatment with RNAse A and Pepsin for 15 min each at 37 °C. Slides were then re-fixed and then dehydrated with 70%, 90% and 100% ethanol. Telomere FISH was performed as above, and a centromeric CENPB probe was used in addition to the telomere probe (PNABio). Single telomere signal intensities were measured using the Nikon NIS Elements AR software. Each chromosome was isolated as single ROI, and telomeres were selected as objects creating a binary layer based on their intensities. The binary object intensities were then collected and separately binned into intervals of 1000 to obtain histograms using PRISM. Quantification of telomeric signal-free ends and fragile telomeres per metaphase was normalized to chromosome number.

## Statistics and reproducibility
The number of biological and technical replicates are noted in all Fig. legends and methods. All statistical analysis was performed in GraphPad Prism 10. No statistical method was used to predetermine sample size. Investigators were not blinded to allocation during experiments and outcome assessments. No statistical method was used to predetermine sample size. No data was excluded from analyses. The experiments were not randomized. The investigators were not blinded to allocation during experiments and outcome assessment.

## Contact for reagent and resource sharing
Further information and requests for reagents should be directed to and will be fulfilled by the Lead Contact, Patricia L. Opresko (plo4@pitt.edu).

## Reporting summary
Further information on research design is available in the Nature Portfolio Reporting Summary linked to this article.

## Data availability
The data generated in this study are provided in the Supplementary Information and Source Data files. All relevant data are available from the authors upon reasonable request. Source data are provided in this paper.

## Code availability
Matlab scripts were uploaded to Zenodo with the following https://doi.org/10.5281/zenodo.17364085.

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

## Acknowledgements

We are grateful to Ariana Detwiler (UPMC Hillman Cancer Center) for technical support and Dr. Ryan Barnes (Kansas University Medical Center) for critical feedback. We thank Dr. Ben Van Houten (University of Pittsburgh) for advice on the C-Trap experimental setup. This research was supported by NIH grants F32CA275287 (to S.L.S) T32GM144273 and F30DK135340 (to W.M.), R35GM147238 (to M.H.), R01DK107716 (to. S.A.), F32ES034982 (to M.A.S.), R35ES030396 (to P.L.O), R01CA207342 (to P.L.O. and S.M.), R01GM095850 and R35GM153235 (to M.D.S). This project used facilities at the UPMC Hillman Cancer Center, which is supported in part by award P30CA047904 and S10OD032158.

## Author contributions

S.L.S. and P.L.O. conceived the study and designed the biochemistry and cellular experiments, with assistance from S.M. S.L.S. conducted the telomerase biochemical activity and binding experiments and cellular experiments to examine new telomeric repeat synthesis, cell survival, and telomere integrity after 6-thio-dG treatment. M.G. assisted S.L.S. with experiments. M.B. and M.D.S. conducted smFRET experiments and analyses. W.M., N.L. and S.A. performed and analyzed data for the Super-telomerase telomere lengthening experiments. R.D. and M.H. designed, conducted, and analyzed data for the DNA polymerase δ reactions with 6-thio-dGTP. A.H. and J.K.A. prepared and provided lentivirus for WT and C50/56 A hTR expression and assisted with analysis of variant repeat addition. M.A.S. assisted S.L.S. with design, performance and analysis of the C-TRAP experiments and conducted the structural modeling of telomerase with 6-thio-dG in the active site. S.L.S. and P.L.O. wrote the manuscript with assistance from the other authors.

## Competing interests

The authors declare no competing interests.

## Additional information

[1]UPMC Hillman Cancer Center, Pittsburgh, PA, USA. [2]Department of Chemistry and Biochemistry, University of California, Santa Cruz, Santa Cruz, CA, USA. [3]Division of Hematology/Oncology and Stem Cell Program, Boston Children's Hospital, Boston, MA, USA. [4]Pediatric Oncology, Dana-Farber Cancer Institute, Boston, MA, USA. [5]Biological and Biomedical Sciences Program, Harvard/MIT MD-PhD Program, Harvard Stem Cell Institute, Harvard Initiative for RNA Medicine, and Department of Pediatrics, Harvard Medical School, Boston, MA, USA. [6]Department of Chemistry, The Pennsylvania State University, University Park, PA, USA. [7]Dorothy P. and Richard P. Simmons Center for Interstitial Lung Disease, University of Pittsburgh, Pittsburgh, PA, USA. [8]Division of Pulmonary, Allergy, Critical Care, and Sleep Medicine, University of Pittsburgh, Pittsburgh, PA, USA. [9]Department of Pharmacology and Chemical Biology, University of Pittsburgh School of Medicine, Pittsburgh, PA, USA. [10]Program in Cell, Molecular, Developmental Biology and Biophysics, Johns Hopkins University, Baltimore, MD, USA. [11]Program in Cellular and Molecular Medicine, Boston Children's Hospital, Harvard Medical School, Boston, MA 02115, USA. [12]RNA Center, University of California Santa Cruz, Santa Cruz, CA, USA. [13]Department of Environmental and Occupational Health, University of Pittsburgh School of Public Health, Pittsburgh, PA, USA. ✉e-mail: plo4@pitt.edu

