## [Transparent Peer Review file · Nature Communications]

Therapeutic 6-thio-deoxyguanosine inhibits telomere elongation in cancer cells by inducing a non-productive stalled telomerase complex

Corresponding Author: Dr Patricia Opresko

Version 1:

Reviewer comments:

Reviewer #1

(Remarks to the Author)

The submitted manuscript titled “Therapeutic 6-thio-deoxyguanosine inhibits telomere elongation in cancer cells by inducing a non-productive stalled telomerase complex” describes the molecular mechanism of how 6-thio-dGTP inhibits telomere synthesis by telomerase. Defining the mechanism of 6-thio-dGTP-mediated telomerase inhibition is important, as it serves as a basis for developing other nucleotide analog-based telomerase inhibitors, thereby advancing potential strategies for targeting telomerase in cancer treatment.

The authors carried out multiple ensemble and single-molecule techniques to rigorously test the telomerase action in telomere synthesis in vitro. Their results demonstrate that telomerase can effectively substitute normal dGTP for 6-thio-dGTP and incorporate it into the newly synthesized DNA, which substantially reduces the processivity of telomerase. Their observations also suggest that 6-thio-dGTP does not inhibit telomerase binding to ssDNA, but its incorporation at the final nucleotide position inhibits further elongation. To support the relevance of their observations in cells, the authors demonstrate that 6-thio-dGTP exposure reduces variant telomeric repeat addition by a mutant telomerase RNA and reduces the proliferation of cancer cells with very short telomeres. In total, the authors propose that 6-thio-dGTP could specifically target cancer cells with very short telomeres due to its effects on telomerase. To further support their conclusions the authors could carry out additional analyses and improve their data presentation by addressing the specific points below.

Major Comments.

1. The single-molecule FRET analysis could be improved by additional analysis. A major conclusion from these experiments is 6-thio-dGTP does not induce substrate dissociation, which is supported by a similar number of molecules observed in each condition. Since this is a critical conclusion, it has to be supported by a more detailed analysis. In fact, while the starting number of molecules is identical, 6-thio-dGTP exposure does reduced the number of molecules at each time point by at least 25%. This is a substantial reduction, especially considering that a significant number of the observed signals by the authors own admission could be inactive telomerase RNPs. The fraction of inactive molecules could be measured by determining the fraction of molecules that do not shift to the low FRET state.
2. In their analysis of telomere elongation in cells the single channel images should be presented in grey scale, rather than false color. It is very challenging to assess the images. The authors quantify the number and intensity of the variant repeats, which represent newly added DNA. In the images presented it also appears that the number and intensity of the TTAGGG repeat signal is lowered by 6-thio-dGTP. Since these should represent the bulk of telomeric DNA it should not be altered significantly by a short exposure to 6-thio-dGTP. It would be valuable for the authors to assess the number and intensity of TTAGGG repeats to rule out other impacts on telomeres, for example 6-thio-dGTP incorporation during telomere replication.
3. The C-trap experiments are unfortunately not very convincing or insufficiently explained. This approach could be very powerful to analyze telomerase binding to telomeric DNA. It is unclear what the signal is the authors are analyzing in the image shown. It appears as if the red signal TERT signal is covering the entire dsDNA handle. Is there perpendicular flow applied to the trapped DNA to spatially separate the ssDNA from the dsDNA? How do the authors confirm the binding is specific and requires base pairing? Why would TERT also associate with the dsDNA? Where is the fluorescein fiducial mark in the image shown? The authors could consider removing these experiments, since they are supplemental. Alternative additional controls are required to demonstrate the observed signal are mediated by base pairing mediated

interactions of TR with the ssDNA overhang.

4. It is unclear how the sensitivity of the HeLa VST compared to HeLa LT was determined. The graphs show stars indicating significant differences. Were those assessed relative to the control or HeLa VST vs HeLa LT. The authors conclude that HeLa VST is more sensitive than HeLa LT and U2OS. Is the cell survival at 2.5 μ M 6-thio-dGTP statistically different between the 3 cell lines?

Minor Points.

1. The POT1/TPP1 stimulation shown in figure 1 does not appear to be very robust and there is no statistics provided for any of the data presented. Was this critical experiment carried out multiple times?
2. Figure 3C 1. Error bars are not lined up with the bar graph. What is the N and is it mean and SD?
3. The authors cite Schmidt et al. 2016 to support Halo-TERT being active. Is the construct used in this study the same? Schmidt et al used 3xFLAG-Halo-TERT, based on the nomenclature the tags are ordered differently here.
4. In Figure 3D it would be helpful to show the TERT signal alone in grey scale to appreciate the binding. TERT in the well could be TERT alone without TR, since TERT is always present in large excess to TR when overexpressed in HEK cells.
5. Figures 4 and 5 could be combined into one.
6. It is established, that newly synthesized telomere DNA can dynamically fold into G-quadruplexes (G4), which modulates its interaction with telomerase anchor site. Here, authors did not discuss how 6-thio-dGTP can alter potential telomeric G4 structures in this scenario.
7. In the Fig. 7f, the statistical significance looks questionable. Authors state in the text that the fraction of dysfunctional DDR+ telomeres increased in both (VST, LT) cell lines, but to a greater extent in HeLa VST. However, according to the figure, the more statistically significant looks HeLa LT (based on the mean value and errors). This is possibly due to the difference in sample size used for the analysis.
8. Fig. 6a – Mistake in the label: Lentiviral transfection. It should be transduction.
9. In the 4th result section, there is missing "1" in POT1. While this variant telomerase is less processive than wild-type, it is stimulated by POT and TPP1 46 and inhibited by 6-thio-dGTP similar to WT telomerase (Extended Data Fig. 4a-c).

Reviewer #2

(Remarks to the Author)

The manuscript by Sanford and colleagues represents a significant advancement in our understanding of the mechanism of action of 6-thio-dG in targeting telomere synthesis. The advancement is possible due to the use of complementary biochemical and biophysical approaches in cells rather than in vitro. The conclusions that 6-thio-dG does not inhibit binding of telomerase to telomeres and induces a non-productive complex rather than enzyme dissociation is strongly supported by the results.

I have a few minor questions.

In experiments in Figure 6, I understand the authors want to exclude the possibility that incorporation of 6-thio-dG is due to reduction in new telomeric repeat synthesis and not reduced growth or senescence. However, ultimately, the goal is to cause reduced growth or senescence in a telomerase- or telomere-length dependent manner. Can the authors comment on the doses or time that would be required for this?

In experiments with RPE cells, what is the explanation for these cells being similarly sensitive to 6-thio-dG with or without hTERT? If the effect of 6-thio-dG is telomerase-dependent, these cells should be sensitive upon expression of hTERT, especially if the telomeres in these cells are short, as the authors show that cells with short telomeres are more sensitive to 6-thio-dG. Is it because these cells do not divide as fast as cancer cells? The authors refer to non-disease cell lines, what are these, cells that don't express telomerase and do not divide rapidly?

The authors use 293 cells lacking hTR, is there a reference for these cells?

Reviewer #3

(Remarks to the Author)

This manuscript presents a detailed investigation into the mechanism and therapeutic potential of 6-thio-2'-deoxyguanosine (6-thio-dG), a telomerase inhibitor, in telomerase-positive cancer cells. First, the authors present a series of biochemical and single molecule FRET studies to try to establish the mechanism of 6-thio-dGTP inhibition of telomerase activity in vitro. They show that, not surprisingly, 6-thio-dGTP can be incorporated into DNA synthesized by telomerase; however, it inhibits the synthesis of multiple telomere repeats. The authors conclude that 6-thio-dG can be readily incorporated into the first telomere repeat but that it inhibits the translocation step that gives rise to repeat addition processivity (RAP). Second, the authors show that treatment with 6-thio-dG in cellular studies results in shortened telomeres and that cells with short telomeres are especially sensitive to 6-thio-dG. The cellular studies are compelling. The in vitro studies expand on previous work from the same lab on 6-thio-dG as a potential strategy for targeting telomerase in cancer. Some aspects of the assays and FRET studies were difficult to understand, as detailed in the comments below. Overall this is an interesting and potentially important study of the mechanism and actions of a potential telomerase therapeutic.

(1) In Fig. 1, the authors use 5 nM DNA primer together with 500 nM POT1 and 500 nM TPP1 in their rescue experiment.

Since POT1 specifically binds to single-stranded G-rich DNA, using such a high concentration relative to the DNA substrate may introduce unintended artifacts, such as reduced activity (actually visible in the Figure) due to sequestration of the primer by POT1 binding. To improve the reliability of this experiment, the authors should optimize the relative concentrations of the primer, POT1, and TPP1. Previous studies (Sekne et al., *Science*, 2022; Liu et al., *Nature*, 2022; Nandakumar et al *Nature* 2012) provide useful information for titrating POT1–TPP1 concentrations in activity assays. Also for Figure 1, please clarify the identity of the band that appears below the primer band.

(2) A major conclusion of this paper on the mechanism of 6-thio-dG inhibition is that it inhibits the translocation step. That is, the nucleotides for a single telomere repeat can be readily incorporated, but subsequent telomere repeats cannot (or are inhibited from) being added, although the primer DNA remains bound to telomerase. Previous work by the authors (Sanford et al, *Nat Comm*, 2020) showed activity assays (Figure 4 b-d) that clearly show inhibition of nucleotide repeat addition (NAP), i.e. there are strong bands for each 6-thio-G addition (indicating that synthesis stopped) for the first repeat and beyond. Similar data is present in Figure 1 of this paper at the highest 6dG concentrations (100 M, 20X dGTP). Specifically, there are strong bands at the end of the first telomere repeat and then for a second telomere repeat, with especially strong bands where there are (likely) 3 consecutive 6dG incorporated, which would be expected to be highly destabilizing for the duplex. Thus, it seems that it might not be translocation but the increasing number of 6dG in a row that inhibits synthesis. Can this be clarified? Also, please address whether just RAP or NAP and RAP are affected (see next point).

(3) In another set of clever assays (Figure 3), a single (or 2) 6dG was added at different positions that would pair with the alignment region and/or beginning of template, and then activity assays with dNTPs were done. The authors quantify the % of primer extension and find that (except for one that would pair with the 3'-end), the % of primers with 6dG that were extended was about half of that for primers with dG. Because the primers are in excess, in this experiment maximally 20% of primers were extended. Can the authors clarify what they think is happening here? Do some enzymes manage to overcome the stalling and then normal synthesis takes place? Is the lower RAP just due to a slower enzyme rate when 6dG is incorporated? I recommend that the authors show a time course for telomerase assays to distinguish whether the apparent decrease in processivity is due to decrease in rate of enzyme when 6-thio-dG is incorporated.

(4) Several things need clarifying in Figure 3. (a) In Fig.3a, the figure caption needs to specify that the numbers under the lanes refer to the panel c. (b) In Figure 3d, what are the extra (slower migrating) bands for the unbound DNA that occur as telomerase concentration increases? (c) Please explain how the free vs bound DNA is quantitated. (d) How can the telomerase concentration be quantitated if there is aggregation at the top of the gel at increasing concentrations of telomerase? (e) The figure legend and methods say 1-20 nM Halo-telomerase was used, but the Figure 3 panels d and e indicate 1-80 nM. Please correct. (f) In the related extended data Figure 3, that does appear to go only from 0 to 20 nM telomerase, please quantitate the results for the DNA "scramble control" substrate. The unbound DNA (green) appears to decrease in intensity. For both gels, why does the telomerase look like two different bands? (g) Figure 3e should use different colors from Figure 3d.

(5) In describing the results of the single molecule FRET experiments, the authors state (line 215-216) "This is consistent with a reduction of processive telomere DNA repeat synthesis as shown in our bulk primer extension assays under the same conditions (Fig 1 b-c and 3a-c). This statement is incorrect. The conditions for experiments in Fig 1, Fig 3, and Fig 4 are all different. For Fig 1, all 4 regular dNTPs plus increasing amounts of 6-thio-dG are used; for Fig 3, one (or two) 6-thio-dG are incorporated into the primers, then only regular dNTPs are used; for the FRET experiments in Fig 4 and 5, dTTP, dATP, and 6-thio-dGTP (and no dGTP) were used and at different concentrations.

For the single molecule experiments, it is important to emphasize that no dGTP is added, as this means that every place where G occurs it will be 6-thio-dG. Otherwise it is easy to be confused by this experiment.

(5a) Also, can the author's correlate the peak positions in the FRET histograms with the approximate number of telomere repeats synthesized? This information would be helpful in interpreting these experiments.

(6) In Fig. 2, The authors used DNA polymerase δ , a high-fidelity enzyme with an error rate of approximately 1 in 10^7 base pairs, to assess the drug's specificity, and find "minor" effect of 6-thio-dGTP on the enzyme's activity (processivity). However, the fact that it does decrease processivity significantly (% of full length products) does seem to me to be relevant to its stronger effect on the short template of telomerase in terms of thinking about mechanism.

Based on this assay, the authors conclude that telomerase is specifically inhibited by 6-thio-dGTP (vs other polymerases). However, it remains unclear whether similar results would be observed with lower-fidelity DNA polymerases. The authors should provide assay results using a lower-fidelity DNA polymerase as well (or explain why this is not needed). The authors conclude (lines 328-329) that "6-thio-dGTP is a selective telomerase inhibitor, since it does not significantly impair replicative DNA polymerase δ processivity"; however, I do not see how this one example conclusively applies to all DNA polymerases.

Note also, lines 296-298: "Interestingly, non-diseased retinal pigment epithelial (RPE) cells with and without hTERT expression exhibited similar 6-thio-dG sensitivity, and were less sensitive than the cancer cell lines (Extended Data Fig. 6a). These data suggest telomerase status alone does not determine sensitivity to 6-thio-G..." also seems to indicate some lack of specificity. Please explain.

(7) The authors show some data that indicates that 6-thio-dG shows less toxicity to telomerase-deficient, non-tumor cells and telomerase active cells (Extended data Figure 6). However, if 6-thio-dGTP can be readily incorporated into DNA

polymerases, couldn't this also cause a toxic effect on cells due to misincorporation of 6-thio-dGTP opposite T, as 6-thio-G-T mismatches are more stable than G-T mismatches (Bohon and de los Santos Nuc Acids Research 2005)?

(8) In Fig.7, A non-cancer cell line should be included as a control to evaluate potential toxicity in normal cells in this figure, not just referenced to in Extended Data Figure 6.

(9) According to the figure caption, the models in Figure 8 b-e were generated from PDB 7BG9 (human telomerase structure with DNA by Ghanim et al Nature 2021) and 6USR (*Tribolium castaneum* TERT). Why was the *Tribolium* structure used at all? (Actually it is not clear from the figure that it was used.) Also, why not use the most recent (higher resolution and more accurate) structures in PDB for modeling?

Minor points

(10) The authors state in line 51, "Human telomeres consist of approximately 10–15 kilobases (kb) of tandem double-stranded GGTTAG repeats," but do not mention the typical length of the 3' single-stranded (ss) G-rich overhang. Including an approximate range would provide a more complete overview of telomere structure.

(11) The statement in line 103, "After incorporating an incoming dNTP, the active site moves to position the next template base," is not accurate. The active site itself remains fixed; rather, the short RNA–DNA duplex translocates to make the active site vacant for the next nucleotide incorporation.

(12) There is an inconsistency between the primer-template alignment base pairing in the mechanism illustrated in Figures 1, 3, and 4, i.e. the schematic in Fig 4 differs from the others.

(13) Line 212 mentions a 60-minute time point for the FRET experiments in Figure 4, but the longest time point shown is 30 minutes.

(14) Line 524 in Methods: "smFRET Telomerase reactions. For telomerase reactions, (5 µl) U42-LD555 labeled telomerase (3µl) were mixed with a 1 nM biotinylated Cy5-labeled smFRET primer ..." What is the concentration of telomerase?

(15) Please check methods and other text carefully, as I found many small errors, e.g. line 616 mentions "ultracentrifuged at 13k rpm for one minute."; Extended Data Fig 2. "pol d"; and in the presence of 6-thio-dGTP or dGTP. Extended Data Fig 4. "(k-m)".

(16) Throughout the manuscript, the authors refer to "telomerase binding" and "dissociation". In most cases they are referring to DNA primer binding to telomerase or dissociation of primer from telomerase. For example, line 191-192, "Collectively, these results confirm that a pre-existing 6-thio-dG does not impair telomerase binding". Please be specific throughout that what is meant, e.g. "...a pre-existing 6-thio-dG in the DNA primer does not impair primer binding to telomerase" (suggested changes in italics).

(17) Figure 4, please label horizontal axes with numbers throughout. Please add an arrow to the position of the mid-peak. Figure 5b, please add tick marks next to numbers. Figure 5a, add the vertical scale to the other side as well.

(18) Methods for extended data Figure 2b is not described anywhere.

Reviewer #4

(Remarks to the Author)

Version 2:

Reviewer comments:

Reviewer #1

(Remarks to the Author)

The authors have included extensive additional experimentation and analysis and have fully addressed the concerns raised in my review of the original manuscript. Congratulations on an important contribution to the telomerase field.

Reviewer #2

(Remarks to the Author)

The revised version of the manuscript addresses my concerns. Perhaps some of the response to comment 1 could be incorporated into the text.

Reviewer #3

(Remarks to the Author)

The authors have substantially addressed our criticisms and suggestions to improve the manuscript. In my opinion, this excellent work is now ready for acceptance.

Reviewer #4

(Remarks to the Author)

We thank the reviewers for their helpful comments and critiques. We have addressed each as detailed below, and believe that new experiments and clarification of the text have greatly strengthened the manuscript. We have also corrected “Extended Data Fig.” to “Supplementary Fig.” to comply with the *Nature Communications* style.

Reviewer #1 - Telomeres, structural bio (Remarks to the Author):

The submitted manuscript titled “Therapeutic 6-thio-deoxyguanosine inhibits telomere elongation in cancer cells by inducing a non-productive stalled telomerase complex” describes the molecular mechanism of how 6-thio-dGTP inhibits telomere synthesis by telomerase. Defining the mechanism of 6-thio-dGTP-mediated telomerase inhibition is important, as it serves as a basis for developing other nucleotide analog-based telomerase inhibitors, thereby advancing potential strategies for targeting telomerase in cancer treatment.

The authors carried out multiple ensemble and single-molecule techniques to rigorously test the telomerase action in telomere synthesis in vitro. Their results demonstrate that telomerase can effectively substitute normal dGTP for 6-thio-dGTP and incorporate it into the newly synthesized DNA, which substantially reduces the processivity of telomerase. Their observations also suggest that 6-thio-dGTP does not inhibit telomerase binding to ssDNA, but its incorporation at the final nucleotide position inhibits further elongation. To support the relevance of their observations in cells, the authors demonstrate that 6-thio-dGTP exposure reduces variant telomeric repeat addition by a mutant telomerase RNA and reduces the proliferation of cancer cells with very short telomeres. In total, the authors propose that 6-thio-dGTP could specifically target cancer cells with very short telomeres due to its effects on telomerase. To further support their conclusions the authors could carry out additional analyses and improve their data presentation by addressing the specific points below.

Major Comments.

1. The single-molecule FRET analysis could be improved by additional analysis. A major conclusion from these experiments is 6-thio-dGTP does not induce substrate dissociation, which is supported by a similar number of molecules observed in each condition. Since this is a critical conclusion, it has to be supported by a more detailed analysis. In fact, while the starting number of molecules is identical, 6-thio-dGTP exposure does reduced the number of molecules at each time point by at least 25%. This is a substantial reduction, especially considering that a significant number of the observed signals by the authors own admission could be inactive telomerase RNPs. The fraction of inactive molecules could be measured by determining the fraction of molecules that do not shift to the low FRET state.

Thank you for this comment. We agree that, in principle, tracking the number of telomerase-DNA complexes on the slides surface as a function of time could report on the stability of the complex. We tried repeating the smFRET experiments taking a slightly different approach than what was presented in the initial submission. The main distinction is that we took individual movies of a single field of view at each indicated time point. The rationale for this approach was to improve the precision of the time points being presented. In the initial submission, we collected twenty unique fields of view, analyzed all molecules in these fields over a two second window, and generated the smFRET histogram from those data (now shown in Supplementary Fig. 4). In the present approach, we take just a single field of view for each time point, analyze each trace manually to correct for any background adjustments, and to ensure that all traces show single step photobleaching for both the donor and acceptor dyes. Then histograms are

generated using the first several seconds of data from each corrected smFRET trajectory (now shown in Figs. 4 and 5).

Collecting and analyzing the data in this manner has the advantage of correcting the background for every molecule, but also has the tradeoff of typically generating a few hundred molecules per time point. Ultimately, irrespective of the data collection approach, we found that the field to field and day to day variation in the glass slide and sample deposition efficiency is too noisy to make quantitative conclusions about the binding efficiency of the telomerase-DNA complexes in each of the experimental conditions.

Importantly, we found that our conclusion from the initial submission stands – namely, that 6-thio-dGTP induces dynamics in the telomerase-DNA complex but does not promote the drop in FRET observed with standard dNTPs. We have revised the writing to remove emphasis from the smFRET measurements regarding dissociation kinetics and rather use the other data (EMSAs and the C-trap measurements) to support claims about binding stability under different experimental conditions.

The major conclusions of the smFRET data remain the same – that only standard dNTPs support the drop in observed FRET values as a function of time, a signal that has been previously reported by our lab to correspond to processive telomere DNA synthesis (Parks and Stone, Nature Communications) ¹. In contrast, the conditions that include 6-thio-dG are distinct from the negative controls but suggest that the small amount of smFRET dynamics observed likely reflects completion of one telomere DNA repeat but fails to stably translocate and re-prime the next round of processive repeat synthesis as observed in the ensemble biochemical primer extension assays. We decided to include the data from both collection methods for robustness (Figs. 4,5 and Supplementary Fig. 4).

2. In their analysis of telomere elongation in cells the single channel images should be presented in grey scale, rather than false color. It is very challenging to assess the images.

Done. We included this in Supplementary Fig. 5d.

The authors quantify the number and intensity of the variant repeats, which represent newly added DNA. In the images presented it also appears that the number and intensity of the TTAGGG repeat signal is lowered by 6-thio-dGTP. Since these should represent the bulk of telomeric DNA it should not be altered significantly by a short exposure to 6-thio-dGTP. It would be valuable for the authors to assess the number and intensity of TTAGGG repeats to rule out other impacts on telomeres, for example 6-thio-dGTP incorporation during telomere replication. Excellent suggestion. We now show the quantification of the wild type TTAGGG repeats in Supplementary Fig. 5e (formerly Extended Data 4). The reviewer is correct that we also see a reduction in TTAGGG repeats at the higher 6-thio-dG treatments. However, there is no significant difference between 1 and 2.5 μ M 6-thio-dG for wild type TTAGGG repeats, unlike for the variant TTAGGT sequencing marking new repeat addition. Notably, these cells express both WT hTR and variant hTR, therefore, the reduction in TTAGGG repeats could be due to reduced WT telomerase activity, but also impaired telomere replication. Consistent with this, we observed that 6-thio-dG increases telomere fragility (Fig. 7d). Therefore, we cannot rule out that 6-thio-dGTP incorporation during telomere replication may impact telomere stability and length. Furthermore, consistent with the 72 h treatment with 6-thio-dG (Supplementary Fig. 5e) we also observed decreased TTAGGG sum intensity after even a very low dose of 6-thio-dG (0.1 μ M) for 14 days (Fig. 7g). However, as a control, we show that there is no reduction in signal intensity of the centromeric PNA probe after 6-thio-dG treatment in Supplementary Data Fig. 5k (formerly Extended Data Fig. 4i), underscoring the telomere specific effects.

3. The C-trap experiments are unfortunately not very convincing or insufficiently explained. This approach could be very powerful to analyze telomerase binding to telomeric DNA. It is unclear what the signal is the authors are analyzing in the image shown. It appears as if the red signal TERT signal is covering the entire dsDNA handle. Is there perpendicular flow applied to the trapped DNA to spatially separate the ssDNA from the dsDNA? How do the authors confirm the binding is specific and requires base pairing? Why would TERT also associate with the dsDNA? Where is the fluorescein fiducial mark in the image shown? The authors could consider removing these experiments, since they are supplemental. Alternative additional controls are required to demonstrate the observed signal are mediated by base pairing mediated interactions of TR with the ssDNA overhang.

We have addressed each question as follows: 1) The red line is a time course trajectory of telomerase binding to a single region on the DNA tightrope. Therefore, this line does not represent telomerase binding to the full length of the DNA tight rope. The experiment is not conducted under flow, so movement of the telomeric overhang is not restricted and likely quite dynamic. Although the fluorescein label on the overhang is much less photostable than the HaloTag ligand (and typically photobleaches within 5 seconds of exposure), we now used selective excitation to collect new kymographs that show direct colocalization of the red HaloTag-TERT with the blue fluorescein signal (see Supplementary Fig. 3d). This new data more clearly shows that TERT preferentially binds the telomeric overhang, although some transient dsDNA samplings were also observed. 2) We confirmed that telomerase binding is base-pairing specific by including new experiments to show that telomerase does not bind to a non-telomeric overhang. Over the course of 5 minute collections on at least 10 DNA tethers containing a scramble DNA sequence, no TERT binding events > 1 s were observed, clarifying that the long-lived events observed are dependent on base-pairing interactions between the TR and the ssDNA (see Supplementary Fig. 3g). 3) We did not observe evidence of stable telomerase interaction with the dsDNA regions of the tight rope. The brief sampling of dsDNA interactions may represent part of the search process for ssDNA, or may be driven by other dsDNA associating factors known to associate with telomerase such as histones H2A/H2B 4) Since we were able to include additional controls of a non-telomeric overhang and fiducial mark for the overhang, we have opted to retain these supplementary experiments in the manuscript. We thought about adding RNase to degrade the hTR component as a control, but were concerned about effects of any residual RNase, particularly on a shared core instrument.

4. It is unclear how the sensitivity of the HeLa VST compared to HeLa LT was determined. The graphs show stars indicating significant differences. Were those assessed relative to the control or HeLa VST vs HeLa LT. The authors conclude that HeLa VST is more sensitive than HeLa LT and U2OS. Is the cell survival at $2.5 \mu\text{M}$ 6-thio-dGTP statistically different between the 3 cell lines?

We now include a more complete statistical comparison between HeLa VST, HeLa LT and U2OS. By two-way ANOVA the value for HeLa VST is statistically different from HeLa LT, but does not reach significance for U2OS. This may be because we obtained fewer replicates for U2OS. However, compared to untreated, the HeLa VST show the greatest fold reduction at 2.3-fold, compared to 1.5-fold for HeLa LT and 1.7-fold for U2OS. We have now more clearly described the result.

Minor Points.

1. The POT1/TPP1 stimulation shown in figure 1 does not appear to be very robust and there is no statistics provided for any of the data presented. Was this critical experiment carried out multiple times?

While published reports of POT1-TPP1 stimulation primarily use reactions with radiolabeled dGTP, our reactions used radiolabeled end-labeled oligonucleotide primers. With ^{32}P -dGTP the longer telomerase products appear more intense because they have a higher number of incorporated radiolabeled dGs than shorter products. For our reactions with end-labeled primers all the products have a single radiolabeled ^{32}P . Therefore, the decrease in intensity of the short products with POT1-TPP1 shows that POT1-TPP1 increases the relative proportion of long products. Similar results are shown in published studies with end-labeled primers (see Fig. 5 from Jansson et al, PNAS, 2019) ². To increase rigor, we have 1) repeated the experiment 3 times to obtain mean \pm s.d. for the IC50 and 2) quantified the proportion of short product (1-4 repeats added) to long product (> 4 repeats added) (Fig. 1b-d, and Supplementary Fig. 1c-d).

2. Figure 3C 1. Error bars are not lined up with the bar graph. What is the N and is it mean and SD?

Thank you. We fixed the alignment and added to the Figure Legend that the “error bars represent the mean \pm sd from three independent experiments”.

3. The authors cite Schmidt et al. 2016 to support Halo-TERT being active. Is the construct used in this study the same? Schmidt et al used 3xFLAG-Halo-TERT, based on the nomenclature the tags are ordered differently here.

We cite Schmidt et al as precedent that the Halo tag does not interfere with telomerase activity, but the reviewer is correct that the order of the tags differ. Our construct is Halo-3xFLAG-TERT, which we clarified in the Results and Methods. We show in Supplementary Fig. 1a (formerly Extended Data Fig. 1a) that our fusion protein, Halo-3xFLAG-TERT, is active.

4. In Figure 3D it would be helpful to show the TERT signal alone in grey scale to appreciate the binding. TERT in the well could be TERT alone without TR, since TERT is always present in large excess to TR when overexpressed in HEK cells.

We added in Supplementary Fig. 3 (a, b, c) images showing hTERT alone in grey scale. The reviewer raises an excellent point, and we now include in the text that hTERT signal in the well may represent hTERT alone without hTR, since hTERT is expressed in excess of hTR in the HEK293 cells.

5. Figures 4 and 5 could be combined into one.

We have combined the original Figures 4 and 5, into the new Supplementary Fig. 4. This data is retained to show robustness and reproducibility. We repeated the smFRET studies using an image collection method that improves precision in timing and background correction. See response to comment 1 above.

6. It is established, that newly synthesized telomere DNA can dynamically fold into G-quadruplexes (G4), which modulates its interaction with telomerase anchor site. Here, authors did not discuss how 6-thio-dGTP can alter potential telomeric G4 structures in this scenario. Outstanding suggestion. We do not favor the model that 6-thio-dGTP incorporation may inhibit telomerase by disrupting G-quadruplex folding in the product DNA for several reasons. 1) we observed inhibition of elongation after insertion of 1 to 2 6-thio-dGTPs when they are still present in the RNA-DNA hybrid of the active site (see Fig 4 of ³). 2) Previous work showed that a single 6-thio-dG does not significantly perturb or destabilize a G4 of the (GGGG)₄ sequence ⁴. However, contradictory results indicate that while a single 6-thio-dG does not significantly destabilize a G4 consisting of GGGG runs ⁴, it can destabilize a G4 consisting of GG runs ⁵. To address this question more directly, we conducted reactions in which we substituted K⁺ cation, which stabilizes G4, with Li⁺ cation which cannot stabilize G4 structure ⁶. We still observe

significant 6-thio-dGTP inhibition of telomerase even under G4 destabilizing conditions of reactions containing LiCl (now included in Supplementary Fig. 1f-g).

7. In the Fig. 7f, the statistical significance looks questionable. Authors state in the text that the fraction of dysfunctional DDR+ telomeres increased in both (VST, LT) cell lines, but to a greater extent in HeLa VST. However, according to the figure, the more statistically significant looks HeLa LT (based on the mean value and errors). This is possibly due to the difference in sample size used for the analysis.

Thank you. We have re-analyzed the statistics and corrected the result. Both HeLa VST and LT show a statistical increase at $P < 0.0001$.

8. Fig. 6a – Mistake in the label: Lentiviral transfection. It should be transduction.

Thank you. We fixed this mistake.

9. In the 4th result section, there is missing “1” in POT1. While this variant telomerase is less processive than wild-type, it is stimulated by POT and TPP1 46 and inhibited by 6-thio-dGTP similar to WT telomerase (Extended Data Fig. 4a-c).

We fixed this error.

Reviewer #2 - telomeres, cancer (Remarks to the Author):

The manuscript by Sanford and colleagues represents a significant advancement in our understanding of the mechanism of action of 6-thio-dG in targeting telomere synthesis. The advancement is possible due to the use of complementary biochemical and biophysical approaches in cells rather than in vitro. The conclusions that 6-thio-dG does not inhibit binding of telomerase to telomeres and induces a non-productive complex rather than enzyme dissociation is strongly supported by the results.

I have a few minor questions.

1. In experiments in Figure 6, I understand the authors want to exclude the possibility that incorporation of 6-thio-dG is due to reduction in new telomeric repeat synthesis and not reduced growth or senescence. However, ultimately, the goal is to cause reduced growth or senescence in a telomerase- or telomere-length dependent manner. Can the authors comment on the doses or time that would be required for this?

In short, we predict that the dose and time required to induce senescence will be influenced by the telomere length and telomerase status, and so would differ depending on these cellular parameters. After treating HCT116 cells, which have short telomeres (~5 kb), with 5 μ M 6-thio-dG for 72 h we observed no increase in senescence. This is important because we wanted to be certain the live cells we imaged for new telomere synthesis were not senescent and therefore, are still competent for telomerase activity. However, we observed a near 50% reduction in relative cell number with 72 h treatment of 5 μ M 6-thio-dG (Fig. 6e) and with 7 days treatment of 1 μ M 6-thio-dG shown in Supplementary Fig. 5h (formerly Extended Data Fig. 4f). This indicates compromised survival at the higher dose or longer exposure. Colony formation assays revealed a near 50% reduction for HeLa VST cells after 9 day treatment with 2.5 μ M 6-thio-dG treatment, while reduction was less for cell lines with long telomeres (HeLa LT and U2OS) (Fig. 7a). Therefore, our studies suggest that doses between 1 to 5 μ M 6-thio-dG cause a significant reduction in growth after a few days of exposure depending on the cell line.

2. In experiments with RPE cells, what is the explanation for these cells being similarly sensitive to 6-thio-dG with or without hTERT? If the effect of 6-thio-dG is telomerase-dependent, these

cells should be sensitive upon expression of hTERT, especially if the telomeres in these cells are short, as the authors show that cells with short telomeres are more sensitive to 6-thio-dG. Is it because these cells do not divide as fast as cancer cells? The authors refer to non-disease cell lines, what are these, cells that don't express telomerase and do not divide rapidly?

RPE and RPE-hTERT cell lines are characterized as non-diseased because they are derived from retinal pigment epithelium and not from diseased tissue, such as a tumor. We were surprised that these cell lines show similar sensitivity to 6-thio-dG. We and others reported that RPE-hTERT telomere lengths range are ~ 8.5 kb, so not as short as some cancer cell lines. While we do not have a clear explanation for why RPE-hTERT cells are not more sensitive than RPE cells to 6-thio-dG, we think this result is important because it reveals that telomerase status alone is insufficient to predict sensitivity to 6-thio-dG. The reviewer may be correct that differences in sensitivity to 6-thio-dG is related to how fast the cells replicate. However, we favor a previous model that suggested cancer cells with short telomeres replicate with a few critically short telomeres that require telomerase activity to lengthen them at each cycle for viability ⁷. In agreement, we found that 6-thio-dG induces more telomeric signal free ends (i.e. losses) in HeLa VST cells compared to HeLa LT cells (Fig. 7d). We added this possible explanation to the Discussion.

The authors use 293 cells lacking hTR, is there a reference for these cells?

Thank you. We have now included the references.

Reviewer #3 - RNA biology (Remarks to the Author):

This manuscript presents a detailed investigation into the mechanism and therapeutic potential of 6-thio-2'-deoxyguanosine (6-thio-dG), a telomerase inhibitor, in telomerase-positive cancer cells. First, the authors present a series of biochemical and single molecule FRET studies to try to establish the mechanism of 6-thio-dGTP inhibition of telomerase activity in vitro. They show that, not surprisingly, 6-thio-dGTP can be incorporated into DNA synthesized by telomerase; however, it inhibits the synthesis of multiple telomere repeats. The authors conclude that 6-thio-dG can be readily incorporated into the first telomere repeat but that it inhibits the translocation step that gives rise to repeat addition processivity (RAP). Second, the authors show that treatment with 6-thio-dG in cellular studies results in shortened telomeres and that cells with short telomeres are especially sensitive to 6-thio-dG. The cellular studies are compelling. The in vitro studies expand on previous work from the same lab on 6-thio-dG as a potential strategy for targeting telomerase in cancer. Some aspects of the assays and FRET studies were difficult to understand, as detailed in the comments below. Overall this is an interesting and potentially important study of the mechanism and actions of a potential telomerase therapeutic.

(1) In Fig. 1, the authors use 5 nM DNA primer together with 500 nM POT1 and 500 nM TPP1 in their rescue experiment. Since POT1 specifically binds to single-stranded G-rich DNA, using such a high concentration relative to the DNA substrate may introduce unintended artifacts, such as reduced activity (actually visible in the Figure) due to sequestration of the primer by POT1 binding. To improve the reliability of this experiment, the authors should optimize the relative concentrations of the primer, POT1, and TPP1. Previous studies (Sekne et al., Science, 2022; Liu et al., Nature, 2022; Nandakumar et al Nature 2012) provide useful information for titrating POT1–TPP1 concentrations in activity assays.

Also for Figure 1, please clarify the identity of the band that appears below the primer band.

Thank you for the references. We are familiar with these outstanding publications on POT1-TPP1 and telomerase. However, a major difference in these studies is that they use radiolabeled dGTP, rather than end-labeled radiolabeled oligonucleotide primers, as in our study. With ³²P-dGTP the longer telomerase products appear more intense because they have

a higher number of incorporated radiolabeled dGs than shorter products. For our reactions with end-labeled primers all the products have a single radiolabeled ^{32}P . Therefore, the decrease in intensity of the short products with POT1-TPP1 shows that POT1-TPP1 increases the relative proportion of long products. To increase rigor, we have 1) repeated the experiment 3 times to obtain mean \pm SD for the IC50 and 2) quantified the proportion of short product (1-4 repeats added) to long product (> 4 repeats added) (Fig. 1b-d, and Supplementary Fig. 1c-d). See also response to reviewer 1.

We use ^{32}P -end labeled primers rather than ^{32}P -dGTP because it allows us to better detect products terminated prior to dGTP addition, and allows detection of incorporated 6-thio-dG (see Fig. 3, Sanford et al, Nat Comm 2020)³. We previously optimized the POT1-TPP1 concentration relative to telomerase. We also use the (TTAGGGTTAGCGTTAGGG) primer A5 (see⁸) to prevent POT1 binding to the 3' end which may occlude telomerase loading. This is now clarified in the Fig. 1b.

Regarding the band below the primer, we note that while the primers are gel purified some impurities remain. You can see this in the gel to the left which represents a reaction with only the primer.

(2) A major conclusion of this paper on the mechanism of 6-thio-dG inhibition is that it inhibits the translocation step. That is, the nucleotides for a single telomere repeat can be readily incorporated, but subsequent telomere repeats cannot (or are inhibited from) being added, although the primer DNA remains bound to telomerase. Previous work by the authors (Sanford et al, Nat Comm, 2020) showed activity assays (Figure 4 b-d) that clearly show inhibition of nucleotide repeat addition (NAP), i.e. there are strong bands for each 6-thio-G addition (indicating that synthesis stopped) for the first repeat and beyond.

The reviewer is correct regarding our previous publication. We observe some termination of synthesis after 6-thio-dGTP insertion, but this is not a block since some synthesis continues. Fig. 3 (middle panel) of the Sanford et al, Nat Comm 2020 manuscript shows that telomerase can insert another 6-thio-dGTP at the 2nd template rC, after inserting a 6-thio-dGTP at the 1st rC. In contrast, telomerase can only insert a single 8-oxo-dGTP, ddITP and AZT-TP, which all block NAP. Fig. 4b and c of Sanford et al, Nat Comm 2020 show that synthesis terminates for about 40-50% of the molecules even after a single 6-thio-dGTP insertion, when translocation is required to insert another 6-thio-dGTP. Finally, Fig. 2a of Sanford et al, Nat Comm 2020, shows that telomerase reactions with dGTP at physiologic concentrations (5 μM) generate products > 20 repeats long, but for reactions with 5 μM 6-thio-dGTP instead, the products were limited to a single repeat, even after addition of only one 6-thio-dGTP. For these reasons, we reached the conclusion in this prior publication that 6-thio-dGTP more strongly inhibits RAP than NAP. The goal of the current manuscript was to determine the mechanism. We reference the moderate effect on NAP that we observed previously in the 3rd paragraph of the Discussion of the current manuscript.

Similar data is present in Figure 1 of this paper at the highest 6dG concentrations (100 mM, 20X dGTP). Specifically, there are strong bands at the end of the first telomere repeat and then for a second telomere repeat, with especially strong bands where there are (likely) 3 consecutive 6dG incorporated, which would be expected to be highly destabilizing for the duplex. Thus, it seems that it might not be translocation but the increasing number of 6dG in a row that inhibits synthesis. Can this be clarified? Also, please address whether just RAP or NAP and RAP are affected (see next point).

For reactions in Fig. 1b we cannot be certain that 6-thio-dGTP (6dG) completely outcompetes dGTP in each reaction, thereby producing a TTA(6dG)(6dG)(6dG) repeat, even with 100 μ M 6-thio-dGTP and 5.2 μ M dGTP. Nevertheless, we observed strong inhibition at 1 and 10 μ M 6-thio-dGTP (~5 fold lower or 2-fold higher than dGTP), and have now included the dNTP concentrations in the figure legend for clarity. However, the reviewer is correct that consecutive 6-thio-dGs are likely more destabilizing, as we show in Figure 3a-b of the current manuscript for oligonucleotide substrates with two 3' 6-thio-dGs. Although it is important to note that we still observe some very long products with this substrate (#7), which is in sharp contrast to reactions containing 10 and 100 μ M 6-thio-dGTP (Fig. 1b). For these reasons, and the reasons detailed in the response above on our published work, we argue our data supports stronger 6-thio-dGTP inhibition of RAP compared to NAP.

(3) In another set of clever assays (Figure 3), a single (or 2) 6dG was added at different positions that would pair with the alignment region and/or beginning of template, and then activity assays with dNTPs were done. The authors quantify the % of primer extension and find that (except for one that would pair with the 3'-end), the % of primers with 6dG that were extended was about half of that for primers with dG. Because the primers are in excess, in this experiment maximally 20% of primers were extended. Can the authors clarify what they think is happening here? Do some enzymes manage to overcome the stalling and then normal synthesis takes place? Is the lower RAP just due to a slower enzyme rate when 6dG is incorporated? I recommend that the authors show a time course for telomerase assays to distinguish whether the apparent decrease in processivity is due to decrease in rate of enzyme when 6-thio-dG is incorporated.

Thank you for this comment. Indeed, we believe our data shows that some enzymes are able to overcome the potential destabilizing effect of a pre-existing 6-thio-dG at the 3' end of an oligonucleotide to extend the telomere. In this context, 6 natural nucleotides are added prior to translocation, and once translocation occurs, the 6-thio-dG would be placed outside the DNA-RNA hybrid region. In contrast, when 6-thio-dGTP is incorporated during the reaction, translocation will place the 6-thio-dG within the newly formed RNA-DNA hybrid, which we propose inhibits re-priming (see model Fig. 8a).

We thank the reviewer for careful consideration of alternative hypotheses to explain 6-thio-dGTP inhibition of telomerase. Our prior work indicates that 6-thio-dGTP does not slow the enzyme rate (Sanford et al, Nat Comm 2020)³. Working with expert enzymologist Bret Freudenthal, we measured the rate of insertion, k_{pol}, using single turnover kinetics. This was done using 6-thio-dGTP and tcTERT, which showed a similar rate of insertion as the natural dGTP with a moderate increase. Importantly, the catalytic efficiency, which is similar to enzyme rate, is largely unchanged.

We are unable to perform single turnover kinetics with the human telomerase complex. However, we now show in a time course experiment, as suggested by the reviewer, that the insertion of a single 6-thio-dGTP is similar to dGTP, as indicated by similar % primer extension over time (Supplementary Fig. 1e). The results agree with our prior kinetic studies using tcTERT.

(4) Several things need clarifying in Figure 3. (a) In Fig.3a, the figure caption needs to specify that the numbers under the lanes refer to the panel c. **Done.** (b) In Figure 3d, what are the extra (slower migrating) bands for the unbound DNA that occur as telomerase concentration increases? We do not know for certain the reason for the slightly higher band and smear of unbound DNA with increasing telomerase concentration. They may arise from impurities of other proteins known to be present in telomerase preparations, including

dyskerin, GAR1, TCAB1, NHP2, as well as histones H2A and H2B. Notably, telomerase is prepared by “pull down” or enrichment from cell extracts (see methods), which is consistent with the published literature. However, since these bands do not colocalize with telomerase, we are confident they represent DNA unbound by telomerase.

(c) Please explain how the free vs bound DNA is quantitated. (d) How can the telomerase concentration be quantitated if there is aggregation at the top of the gel at increasing concentrations of telomerase?

We now clarified how free vs bound DNA was quantitated in the Methods. As reviewer 1 indicated (see response to minor comment #4), not all Flag-hTERT may contain hTR. This was why the standard in the field for quantifying telomerase is to quantify hTR rather than hTERT. Therefore, the hTERT in the wells may represent partially unfolded hTERT or hTERT lacking hTR, and was not included in the quantification. We also now clarify in the figure legends that the graph shows estimated telomerase quantification based on hTR analysis in the preparations. Nevertheless, the graphs clearly show similar amounts of % Bound product for substrates with 6-thio-dG or dG at the 3' end.

(e) The figure legend and methods say 1-20 nM Halo-telomerase was used, but the Figure 3 panels d and e indicate 1-80 nm. Please correct. Thank you for noting this. The figure legend is correct and we now corrected the Figure.

(f) In the related extended data Figure 3, that does appear to go only from 0 to 20 nM telomerase, please quantitate the results for the DNA “scramble control” substrate. The unbound DNA (green) appears to decrease in intensity. For both gels, why does the telomerase look like two different bands? Done. We now show the gels in grey scale as well, and quantified data for the scrambled non-telomeric control substrate (Supplementary Fig. 3a, b, c). Unfortunately, we are unsure why telomerase appears as two bands. However, we speculate that this may be due to some differences in the telomerase holoenzyme conformation which is rather complex and includes hTR, hTERT, dyskerin, GAR1, TCAB1, NHP2, as well as histones H2A and H2B⁹.

(g) Figure 3e should use different colors from Figure 3d. Done.

(5) In describing the results of the single molecule FRET experiments, the authors state (line 215-216) “This is consistent with a reduction of processive telomere DNA repeat synthesis as shown in our bulk primer extension assays under the same conditions (Fig 1 b-c and 3a-c). This statement is incorrect. The conditions for experiments in Fig 1, Fig 3, and Fig 4 are all different. For Fig 1, all 4 regular dNTPs plus increasing amounts of 6-thio-dG are used; for Fig 3, one (or two) 6-thio-dG are incorporated into the primers, then only regular dNTPs are used; for the FRET experiments in Fig 4 and 5, dTTP, dATP, and 6-thio-dGTP (and no dGTP) were used and at different concentrations.

Thank you for this suggestion. We have softened the statement to “similar conditions”. However, to increase rigor, we repeated the smFRET experiments (please see response to Reviewer 1, comment 1) and also added a new experiment in which 100 μ M 6-thio-dGTP is included with the natural dNTPs (dTTP, dATP and dGTP). The results are similar to those obtained when dGTP is omitted, although we observe a slightly larger shifted low FRET peak as expected, since dGTP is now available to compete with 6-thio-dGTP.

The reviewer is correct about the differing conditions between Figs 1 and 3. The goal was to determine how 6-thio-dG impacted telomerase when present as the incoming dNTP versus when present in the primer DNA.

For the single molecule experiments, it is important to emphasize that no dGTP is added, as this means that every place where G occurs it will be 6-thio-dG. Otherwise it is easy to be confused by this experiment. **Done**

(5a) Also, can the author's correlate the peak positions in the FRET histograms with the approximate number of telomere repeats synthesized? This information would be helpful in interpreting these experiments.

Thanks for this excellent suggestion – unfortunately, a direct correlation of specific FRET levels with number of telomere repeats synthesized has thus far not been possible in our assay. We expect the reason for this challenge is that under the conditions of our experiments, the nascent product DNA can exhibit structural dynamics itself – likely as a consequence of G-quadruplex folding and unfolding. We and many others in the field have well-documented the complex structural dynamics of G-quadruplex DNA using smFRET methods and thus we cannot directly relate the FRET signal to the precise length of the DNA product. It is for this reason that we recently developed and reported a novel smFRET based assay that makes use of zero mode waveguide technology to directly track the incorporation kinetics for each successive nucleotide¹⁰. In the future, it will be interesting to explore the impact of 6-thio-dG and other factors on the kinetics of dNTP binding and incorporation in this assay. However, that undertaking is beyond the scope of this manuscript.

(6) In Fig. 2, The authors used DNA polymerase δ , a high-fidelity enzyme with an error rate of approximately 1 in 10^7 base pairs, to assess the drug's specificity, and find “minor” effect of 6-thio-dGTP on the enzyme's activity (processivity). However, the fact that it does decrease processivity significantly (% of full length products) does seem to me to be relevant to its stronger effect on the short template of telomerase in terms of thinking about mechanism.

Based on this assay, the authors conclude that telomerase is specifically inhibited by 6-thio-dGTP (vs other polymerases). However, it remains unclear whether similar results would be observed with lower-fidelity DNA polymerases. The authors should provide assay results using a lower-fidelity DNA polymerase as well (or explain why this is not needed). The authors conclude (lines 328-329) that “6-thio-dGTP is a selective telomerase inhibitor, since it does not significantly impair replicative DNA polymerase δ processivity”; however, I do not see how this one example conclusively applies to all DNA polymerases.

We cite structural work with DNA polymerase β showing that adducted 6-thio-dG is located in the major groove of the DNA helix which is well established to accommodate large adducts to nucleotides. Structures of DNA polymerase β showed 6-thio-dG does not alter the base pairing properties with dC¹¹. We also now cite previous work with DNA polymerase β along with a prior study with DNA polymerases α , γ , and δ ^{11,12}. In addition, we now include a statement that we cannot rule out the possibility that other DNA polymerases may be impacted by 6-thio-dGTP. We also softened our statement to read “Furthermore, we provide evidence that 6-thio-dGTP is likely a selective telomerase inhibitor, since it does not significantly impair replicative DNA polymerase δ processivity”.

Note also, lines 296-298: “Interestingly, non-diseased retinal pigment epithelial (RPE) cells with and without hTERT expression exhibited similar 6-thio-dG sensitivity, and were less sensitive than the cancer cell lines (Extended Data Fig. 6a). These data suggest telomerase status alone does not determine sensitivity to 6-thio-G...” also seems to indicate some lack of specificity. Please explain.

We agree with the reviews, and we think this is an important finding in our study. While our data suggest 6-thio-dGTP may be a selective telomerase inhibitor, it is not “selectively” inserted by telomerase. Replicative DNA polymerases can also insert this therapeutic nucleotide. In the Discussion, we argue telomere length is also an important parameter in determining sensitivity to 6-thio-dGTP. See also response to comment #2 from Reviewer #2. Note Extended Data Fig. 6a is not Supplementary Fig. 7a.

(7) The authors show some data that indicates that 6-thio-dG shows less toxicity to telomerase-deficient, non-tumor cells and telomerase active cells (Extended data Figure 6). However, if 6-thio-dGTP can be readily incorporated into DNA polymerases, couldn't this also cause a toxic effect on cells due to misincorporation of 6-thio-dGTP opposite T, as 6-thio-G-T mismatches are more stable than G-T mismatches (Bohon and de los Santos *Nuc Acids Research* 2005)?

We appreciate this suggestion and are familiar with the study. But we do not currently have any data to indicate that 6-thio-dG treatment leads to 6-thio-dG:T base pairs in the telomeres, and we believe it would be premature to speculate. Previous studies indicate that the mutagenicity of 6-thio-dG is low¹³⁻¹⁵. However, as we note in the Discussion, we propose that 6-thio-dG may impact telomere replication.

(8) In Fig.7, A non-cancer cell line should be included as a control to evaluate potential toxicity in normal cells in this figure, not just referenced to in Extended Data Figure 6.

We chose to place the RPE and RPE-hTERT cells in the Supplement, since non-disease cells have been published previously to be less sensitive to 6-thio-dG^{16,17}.

(9) According to the figure caption, the models in Figure 8 b-e were generated from PDB 7BG9 (human telomerase structure with DNA by Ghanim et al *Nature* 2021) and 6USR (*Tribolium castaneum* TERT). Why was the *Tribolium* structure used at all? (Actually it is not clear from the figure that it was used.) Also, why not use the most recent (higher resolution and more accurate) structures in PDB for modeling?

Thank you for this comment. We have clarified in the figure legend that the cryoEM structure from PDB code 7BG9 was utilized for the global structures in panels **b** and **c**, and the structure from *Tribolium castaneum* TERT used for close-up view of the active site (panels **d** and **e**). The *Tribolium* structure was utilized for these angstrom level measurements because it is the only available structure with an incoming dNTP available, which is of particular interest in this work examining incoming 6-thio-dGTP. Additionally, the precision of the *Tribolium* structures is significantly higher due to the higher resolution than the cryoEM datasets, which is important because the measurements we made are <1 angstroms. Regarding the selection of human cryo-EM structures, we recognize that there have been modest increases in resolution since the initial 2021 study (3.8 vs 3.3 angstrom), but because the global structure does not significantly change at those levels we opted for the 2021 publication for our model because we view it as a major breakthrough in the field of telomerase structural biology.

As evidence, the overlaid telomerase structures are shown below (the 2021 one in green, PDB code 7BG9, and highest resolution one in magenta, PDB code 7QXA). These two structures are largely indistinguishable from a global view, with an RMSD value of 1.0 angstrom.

Minor points

(10) The authors state in line 51, “Human telomeres consist of approximately 10–15 kilobases (kb) of tandem double-stranded GGTTAG repeats,” but do not mention the typical length of the 3' single-stranded (ss) G-rich overhang. Including an approximate range would provide a more complete overview of telomere structure.

Done.

(11) The statement in line 103, “After incorporating an incoming dNTP, the active site moves to position the next template base,” is not accurate. The active site itself remains fixed; rather, the short RNA–DNA duplex translocates to make the active site vacant for the next nucleotide incorporation.

We apologize for the misunderstanding. This sentence describes nucleotide addition processivity prior to translocation to complete the 6 nt repeat. We removed the word “position” for clarity.

(12) There is an inconsistency between the primer-template alignment base pairing in the mechanism illustrated in Figures 1, 3, and 4, i.e. the schematic in Fig 4 differs from the others. We apologize for the confusion. The schematics serve a different purpose for each figure. The schematic in Fig 1 represents a “generic” catalytic cycle of telomerase GGTTAG addition but is not meant to mimic the primer used in 1b. Fig 3b is meant to mimic the primers 1-3 used in Fig 3a. Fig. 4 is meant to show the experimental setup with the primer used in the smFRET experiment. The (TTAGGG)₃ primer is widely used in the telomerase field.

(13) Line 212 mentions a 60-minute time point for the FRET experiments in Figure 4, but the longest time point shown is 30 minutes.

Thank you. We fixed this.

(14) Line 524 in Methods: “smFRET Telomerase reactions. For telomerase reactions, (5 μ l) U42-LD555 labeled telomerase (3 μ l) were mixed with a 1 nM biotinylated Cy5-labeled smFRET primer ...” What is the concentration of telomerase? The standard for telomerase quantification is to conduct RNA dot blots to measure hTR (see Supplementary Fig.1). Our fluorescently

labeled telomerase preps are typically between 1-5 nM. However, for the smFRET experiments we cannot be certain of the concentration of telomerase that is pre-bound to the telomere DNA molecules on the slide. Since our goal is not to quantify binding affinity (K_d), we have not pursued methods to attempt quantification of the pre-bound telomerase concentration. With our experimental setup, we estimate roughly equimolar concentrations of enzyme and DNA (~1 nM), but again we cannot be certain of the concentration of DNA that successfully binds to the slide. Our goal is to measure the internal structural dynamics of preformed and surface-immobilized complexes, which does not require knowledge of precise concentrations.

(15) Please check methods and other text carefully, as I found many small errors, e.g. line 616 mentions “ultracentrifuged at 13k rpm for one minute.”; Extended Data Fig 2. “pol d”; and in the presence of 6-thio-dGTP or dGTP. Extended Data Fig 4. “(k-m)”.

Thank you. We proofread the manuscript more carefully. Extended Data Fig. 2 is now Supplementary Fig. 2, and Extended Data Fig. 4 is now Supplementary Fig. 5.

(16) Throughout the manuscript, the authors refer to “telomerase binding” and “dissociation”. In most cases they are referring to DNA primer binding to telomerase or dissociation of primer from telomerase. For example, line 191-192, “Collectively, these results confirm that a pre-existing 6-thio-dG does not impair telomerase binding”. Please be specific throughout that what is meant, e.g. “...a pre-existing 6-thio-dG in the DNA primer does not impair primer binding to telomerase” (suggested changes in italics).

We improved the precision and clarity of our text.

(17) Figure 4, please label horizontal axes with numbers throughout. Please add an arrow to the position of the mid-peak. Figure 5b, please add tick marks next to numbers. Figure 5a, add the vertical scale to the other side as well.

Thank you. Since the peak is broad with natural dNTPs we removed the phrase “centered-at” from the statement. Stylistically, we chose to add numbers only to the y-axis on the left, since adding to the right as well would be too busy, in our opinion. Tick marks were added to Figure 5b.

(18) Methods for extended data Figure 2b is not described anywhere.

The analysis is described as “the probability of insertion, P_i , for each dNTP insertion step, i , after $i = 1$ was calculated as described previously^{18,19}”.

Reviewer #4 (Remarks to the Author):

Thank you. We appreciate your time and effort.

References

1. Parks, J.W. & Stone, M.D. Coordinated DNA dynamics during the human telomerase catalytic cycle. *Nat Commun* **5**, 4146 (2014).
2. Jansson, L.I. et al. Telomere DNA G-quadruplex folding within actively extending human telomerase. *Proc Natl Acad Sci U S A* **116**, 9350-9359 (2019).
3. Sanford, S.L., Welfer, G.A., Freudenthal, B.D. & Opresko, P.L. Mechanisms of telomerase inhibition by oxidized and therapeutic dNTPs. *Nat Commun* **11**, 5288 (2020).
4. Stefl, R., Spackova, N., Berger, I., Koca, J. & Sponer, J. Molecular dynamics of DNA quadruplex molecules containing inosine, 6-thioguanine and 6-thiopurine. *Biophys J* **80**, 455-68 (2001).
5. Marathias, V.M., Sawicki, M.J. & Bolton, P.H. 6-Thioguanine alters the structure and stability of duplex DNA and inhibits quadruplex DNA formation. *Nucleic Acids Res* **27**, 2860-7 (1999).
6. Lormand, J.D. et al. DNA polymerase delta stalls on telomeric lagging strand templates independently from G-quadruplex formation. *Nucleic Acids Res* **41**, 10323-33 (2013).
7. Zhang, X., Mar, V., Zhou, W., Harrington, L. & Robinson, M.O. Telomere shortening and apoptosis in telomerase-inhibited human tumor cells. *Genes Dev* **13**, 2388-99 (1999).
8. Latrick, C.M. & Cech, T.R. POT1-TPP1 enhances telomerase processivity by slowing primer dissociation and aiding translocation. *EMBO J* **29**, 924-33 (2010).
9. Ghanim, G.E. et al. Structure of human telomerase holoenzyme with bound telomeric DNA. *Nature* **593**, 449-453 (2021).
10. Hentschel, J. et al. Real-time detection of human telomerase DNA synthesis by multiplexed single-molecule FRET. *Biophys J* **122**, 3447-3457 (2023).
11. Schaich, M.A., Smith, M.R., Cloud, A.S., Holloran, S.M. & Freudenthal, B.D. Structures of a DNA Polymerase Inserting Therapeutic Nucleotide Analogues. *Chem Res Toxicol* **30**, 1993-2001 (2017).
12. Ling, Y.H., Nelson, J.A., Cheng, Y.C., Anderson, R.S. & Beattie, K.L. 2'-Deoxy-6-thioguanosine 5'-triphosphate as a substrate for purified human DNA polymerases and calf thymus terminal deoxynucleotidyltransferase in vitro. *Mol Pharmacol* **40**, 508-14 (1991).
13. Swann, P.F. et al. Role of postreplicative DNA mismatch repair in the cytotoxic action of thioguanine. *Science* **273**, 1109-11 (1996).
14. Rappaport, H.P. Replication of the base pair 6-thioguanine/5-methyl-2-pyrimidine with the large Klenow fragment of Escherichia coli DNA polymerase I. *Biochemistry* **32**, 3047-57 (1993).
15. Yuan, B., O'Connor, T.R. & Wang, Y. 6-Thioguanine and S(6)-methylthioguanine are mutagenic in human cells. *ACS Chem Biol* **5**, 1021-7 (2010).
16. Mender, I., Gryaznov, S., Dikmen, Z.G., Wright, W.E. & Shay, J.W. Induction of telomere dysfunction mediated by the telomerase substrate precursor 6-thio-2'-deoxyguanosine. *Cancer Discov* **5**, 82-95 (2015).
17. Mender, I., Gryaznov, S. & Shay, J.W. A novel telomerase substrate precursor rapidly induces telomere dysfunction in telomerase positive cancer cells but not telomerase silent normal cells. *Oncoscience* **2**, 693-5 (2015).
18. Hedglin, M., Pandey, B. & Benkovic, S.J. Stability of the human polymerase delta holoenzyme and its implications in lagging strand DNA synthesis. *Proc Natl Acad Sci U S A* **113**, E1777-86 (2016).
19. Hedglin, M., Pandey, B. & Benkovic, S.J. Characterization of human translesion DNA synthesis across a UV-induced DNA lesion. *Elife* **5**, e19788, 1 - 18 (2016).

We thank the reviewers for their time and effort reviewing our revised manuscript and are delighted that they have no additional concerns.